# LLM-ERM: Sample-Efficient Program Learning via LLM-Guided Search

## Abstract

We seek algorithms for program learning that are both sample-efficient and computationally feasible. In the realizable short-program regime, length-first (Occam/MDL) enumeration achieves near-optimal PAC rates—if the target has a length-$L$ description over alphabet $\Sigma$, finite-class ERM requires only $\mathcal{O}(L \log |\Sigma|/\epsilon)$ samples—but naïve length-first enumeration is computationally infeasible. In contrast, stochastic gradient descent (SGD) is computationally practical yet sample-inefficient. Under the statistical query (SQ) framework, iteration/sample lower bounds scale with SQ dimension, implying exponential data requirements for parities and related families even for short target programs.

To address this gap, we introduce LLM-ERM, a propose-and-verify framework that replaces exhaustive enumeration with an LLM-guided search over candidate programs while retaining ERM-style selection on held-out data. Specifically, we draw $k$ candidates with a pretrained reasoning-augmented LLM, compile and check each on the data, and return the best verified hypothesis, with no feedback, adaptivity, or gradients. Theoretically, we formalize how SQ hardness transfers to SGD iteration complexity on high-SQ-dimension classes. *Empirically, LLM-ERM solves tasks such as parity variants, pattern matching, and primality testing with as few as 200 samples, while SGD-trained transformers overfit even with 100,000 samples*. These results indicate that language-guided program synthesis recovers much of the statistical efficiency of finite-class ERM while remaining computationally tractable, offering a practical route to learning succinct hypotheses beyond the reach of gradient-based training.

## 1 Introduction

At its core, machine learning seeks algorithms that uncover structure in data: given input–output examples, the goal is to recover an unknown function that generalizes to unseen inputs. Classical learning theory provides conditions under which this is possible. In particular, when the target lies in a finite hypothesis class, empirical risk minimization (ERM) requires only a modest number of samples—scaling logarithmically with the class size (Valiant, 1984; Vapnik, 1998). For example, if the target can be expressed as a short program of length $L$ over an alphabet $\Sigma$, then $\mathcal{O}(L \log |\Sigma|)$ samples suffice.

The challenge lies in computation. Exhaustive program enumeration guarantees that we will eventually find the needle, but only by sifting through an exponentially large haystack of candidate programs. Concretely, if the target program has length $L$ over an alphabet $\Sigma$, then the number of candidate strings of length at most $L$ is $|\mathcal{L}_{\leq L}| = \sum_{\ell=1}^{L} |\Sigma|^\ell = \Theta(|\Sigma|^L)$. Even when verifying each candidate requires only linear time in the sample size $m$, the total runtime scales as $\Theta(m |\Sigma|^L)$. In practice, this brute-force search becomes infeasible even for modest $L$ (e.g., $L = 20$ with $|\Sigma| = 10$ already yields $10^{20}$ candidates).

***Modern deep learning flips this trade-off:*** Rather than searching the haystack directly, we train neural networks via stochastic gradient descent (SGD) (Robbins and Monro, 1951; Bottou,

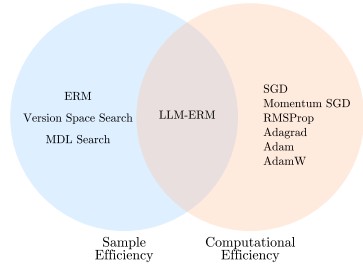

Figure 2: Trade-offs between sample and computational efficiency in program learning. The proposed method (LLM-ERM) lies in the intersection.

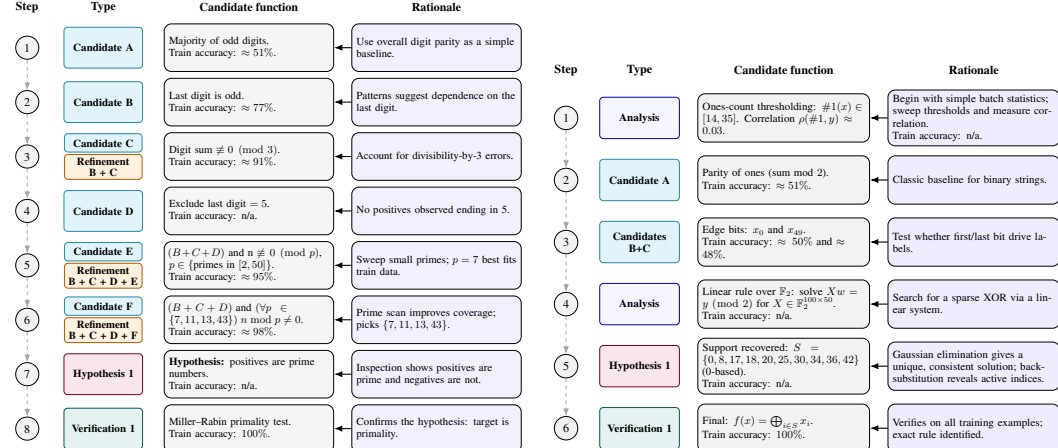

(a) **Reasoning trace for learning primality.** The model starts with digit heuristics, eliminates small-prime multiples, and converges to the Miller–Rabin test.

(b) **Reasoning trace for learning a Random 10-Parity function.** The model starts from simple heuristics, shifts to linear algebra over F2, and ultimately identifies the exact XOR rule over 10 specific indices.

Figure 1: Side-by-side comparison of reasoning traces for two distinct learning tasks. Rules were proposed by GPT-5-Thinking in a single run until convergence. Train accuracy values shown are those the model decided to compute and explicitly include in its reasoning trace. In each subfigure the two columns, **Left:** sequence of proposed rules and **Right:** rationale for each proposal.

2010). Fitting training data in this way is computationally efficient, but it can be provably suboptimal in terms of sample complexity. Viewed through the statistical query (SQ) framework (Kearns, 1998), one finds that SGD may require exponentially many samples on certain high-SQ-dimension families—such as parity or cryptographic-like functions—even though these functions admit succinct program representations. In short, gradient-based methods fail not because the target is deeply hidden, but because their search procedure is poorly matched to the structure of the hypothesis space.

> *Can we design learning algorithms that combine the sample efficiency of finite-class program search with the computational efficiency of modern optimization methods?*

**Contributions.** We revisit program learning through the lens of LLMs and ask whether pretrained reasoning can bridge the gap between statistical and computational efficiency. Our contributions are:

- **Theory: SGD lower bounds via SQ.** We show how statistical-query (SQ) hardness translates into *iteration* complexity for mini-batch SGD. For high SQ-dimension families (e.g., parity), we prove lower bounds where—even if a short correct program exists—gradient-based learners may need exponentially many samples/iterations to reach nontrivial error. In contrast, in the realizable short-program regime, finite-class ERM over programs of length $L$ achieves sample complexity $\mathcal{O}\big(\frac{1}{\epsilon}\big(L \log |\Sigma| + \log(\frac{L}{\delta})\big)\big)$, independent of input dimension, but computationally exponential in $L$.

- **Algorithm: propose–verify LLM-ERM.** We introduce an LLM-guided synthesis procedure that maintains a finite pool of candidate programs and selects among them via empirical risk minimization on held-out data. LLM feedback proposes discrete edits that bias search toward promising regions of program space. The outer loop is gradient-free and non-adaptive w.r.t. validation (early stopping below a threshold), preserving ERM-style generalization while dramatically shrinking the search relative to naive enumeration.

- **Empirics: sample efficiency and cross-dimension generalization.** Across a suite of algorithmic tasks (e.g., parity variants (Full/First-Half/Random-$k$), pattern matching, palindromes, Dyck-2, primality testing, cellular-automata parity, and SHA-256 parity), LLM-ERM typically recovers the *exact* target rule from only 200 labeled examples and generalizes strongly (Figs. 5, 7; Tab. 1). For many cases (e.g., parities, IsPrime), the synthesized programs are *dimension-invariant*, yielding effectively unbounded test accuracy when evaluated beyond the training length (see Tab. 1). On

palindromes and Dyck-2, LLM-ERM discovers high-accuracy but not always exact parsers. In sharp contrast, SGD-trained transformers (e.g., Qwen3-1.7B) fit the training data yet collapse to chance on non-local/recursive tasks, and even scaling to 100k examples fails to fix generalization for Random 10-Parity, Cellular Automata Parity, and digit-restricted IsPrime (Fig. 6). Finally, both approaches remain near chance on `SHA-256` parity, highlighting a high-bar for learning. These trends are robust across model architecture, learning rate, and batch size.

- **Interpretability: the final hypothesis and the learning process are interpretable by construction.** The output is *executable, human-readable code* accompanied by an auditable reasoning trace (Figs. 1, 14 and 15). *This makes both the learned function and the learning process interpretable.* One can inspect intermediate candidates, understand why they were proposed, and validate the final rule mechanistically (e.g., Miller–Rabin for IsPrime, XOR over specific indices for $k$-parities), enabling counterfactual edits and dimension-transfer tests.

## 1.1 RELATED WORK

**PAC learning, Occam's razor and short programs.** We follow the the classical generalization theory, where finite-class ERM has sample complexity $\mathcal{O}(\log|\mathcal{H}|)$ (Valiant, 1984; Vapnik and Chervonenkis, 1971; Vapnik, 1998). The "short program" view instantiates Occam/MDL: a hypothesis encodable in $L$ symbols over alphabet $\Sigma$ admits bounds of order $\mathcal{O}(L\log|\Sigma|)$, up to confidence terms (Blumer et al., 1987; Barron and Cover, 1991; Barron et al., 1998; Rissanen, 1989; McAllester, 1998). The length-first search (Alg. 2) realizes this ERM guarantee but incurs exponential time in description length, reflecting the classic universal-search trade-off (Levin, 1973; Solomonoff, 1964).

**Statistical query (SQ) learning and hardness of learning.** The SQ framework and its refinements (Kearns, 1998; Blum et al., 1994; Feldman, 2017; Reyzin, 2020) yield lower bounds for many concept classes. Parity and related families have large SQ dimension under the uniform distribution, so any SQ learner needs exponentially many (tolerant) queries to achieve nontrivial correlation (Blum et al., 1994; Feldman et al., 2017; Klivans and Sherstov, 2007; Klivans and Kothari, 2014). Intuitively, mini-batch SGD is itself approximately an SQ algorithm: each update averages a bounded statistic over samples (Feldman et al., 2017; 2018; Abbe et al., 2021; Barak et al., 2022). Hence, SQ lower bounds transfer directly to SGD, making its iteration complexity grow with the SQ dimension—exponentially for parities and pseudorandom families under the uniform distribution. Our analysis formalizes this connection, showing how SQ hardness induces exponential sample requirements for gradient-based methods.

**Gradient-based training on algorithmic reasoning.** Beyond worst-case bounds, a long line of work studies when expressive neural families are actually *trainable* with SGD, separating representational power from optimization and sample efficiency (Yehudai and Shamir, 2019; Daniely, 2017). Empirically, SGD-trained neural networks often struggle on parity-like or compositional algorithmic tasks without strong inductive bias or very large data, even when the target is compactly describable (Shalev-Shwartz et al., 2017; Safran and Shamir, 2018; Daniely and Malach, 2020; Barak et al., 2022). The "grokking" phenomenon—delayed generalization after long training on small algorithmic datasets—further highlights the mismatch between the statistical optimum and what SGD discovers (Power et al., 2022). These observations motivate alternatives that retain finite-class guarantees while improving practical search efficiency.

**LLM-guided optimization and evolutionary feedback.** Since the advent of LLMs, researchers have explored ways to elicit task solutions directly via prompting. One line of work is *in-context learning* (ICL), where demonstrations—often with chain-of-thought—induce task procedures without parameter updates (Brown et al., 2020; Min et al., 2022; Wei et al., 2022). Its learning capabilities have been analyzed in a series of papers (Von Oswald et al., 2023; Akyürek et al., 2023; Shen et al., 2024; de Wynter, 2025). Beyond ICL, natural-language or symbolic feedback enables iterative propose–critique–revise loops (e.g., Self-Refine, Reflexion) and even *textual gradients* that treat feedback as a search direction in discrete spaces (Madaan et al., 2023; Shinn et al., 2023; Yuksekgonul et al., 2024). In parallel, evolutionary and neuro-symbolic approaches (SOAR, AlphaEvolve, LEGO) use LLMs to propose edits or modular building blocks, refined via mutation–selection (Pourcel et al., 2025; Novikov et al., 2025; DeepMind, 2025; Bhansali et al., 2024). Within program synthesis, LLMs have been used to generate patches, tests, and rationales that guide iterative repair and verification (Chen et al., 2024; Wang et al., 2024; Hu et al., 2025).

## 2 THEORETICAL ANALYSIS

### 2.1 PROBLEM SETUP

We study *inductive program synthesis* ("program learning"): the target is a binary function $y : \mathcal{X} \to \{\pm 1\}$ implemented by a short program in a fixed language, and the learner receives i.i.d. examples $S = \{(x_i, y(x_i))\}_{i=1}^m$ with $x_i \sim D$. Throughout, we assume the *realizable* setting, i.e., $y \in \mathcal{L}$, where $\mathcal{L}$ is the class of total functions computed by programs in the language (formalized below).

**Language and semantics.** Fix a finite alphabet $\Sigma$ and a programming language $\mathcal{L} \subseteq \Sigma^*$. Each string $u \in \mathcal{L}$ has semantics $[\![u]\!] : \mathcal{X} \rightharpoonup \{\pm 1\}$, a (possibly partial) function that may fail to compile or fail to halt. We write $[\![u]\!](x) = \bot$ when $u$ does not produce an output on $x$. Let $\mathcal{C} := \{ f : \mathcal{X} \to \{\pm 1\} : \exists u \in \mathcal{L} \text{ s.t. } [\![u]\!] \text{ is total and } [\![u]\!] = f \}$. We denote the length of $u$ by $|u|$ (in symbols over $\Sigma$) and write $\mathcal{L}_\ell := \{u \in \mathcal{L} : |u| = \ell\}$. A program is considered total if it defines an output for every input—i.e., it never fails to compile and halts on all $x \in \mathcal{X}$, returning a label in $\{\pm 1\}$.

**Data model and objective.** The learner observes a sample $S = \{(x_i, y_i)\}_{i=1}^m$ with $x_i \overset{\text{i.i.d.}}{\sim} D$ and $y_i = y(x_i)$. For a hypothesis $h : \mathcal{X} \to \{\pm 1\}$, define population error $\mathrm{err}_D(h) := \Pr_{x \sim D}[h(x) \neq y(x)]$ and empirical error $\mathrm{err}_S(h) := \frac{1}{m} \sum_{i=1}^m \mathbf{1}\{h(x_i) \neq y_i\}$. The goal is to output a program $u \in \mathcal{L}$ whose total semantics $[\![u]\!]$ attains small $\mathrm{err}_D$.

**Computational model.** When executing a candidate program $u$ on input $x$, we allow a time budget $T \in \mathbb{N}$ per call; if $u$ fails to compile or does not halt within time $T$, we treat the outcome as $\bot$ and reject $u$ as a hypothesis. This makes search procedures well-defined even when $[\![u]\!]$ is partial.

**Short-program regime.** We will frequently analyze the *short-program* subclass $\mathcal{H}_\ell := \{ [\![u]\!] : u \in \mathcal{L}_\ell, [\![u]\!] \text{ total}\}$, where $\mathcal{H} = \bigcup_{\ell \geq 1} \mathcal{H}_\ell$ and compare (i) explicit search over $\mathcal{H}_\ell$ (finite-class ERM) to (ii) gradient-based learners $h_\theta$ drawn from a proxy hypothesis family $\{h_\theta : \theta \in \Theta\}$.

### 2.2 ANALYZING THE SAMPLE COMPLEXITY

To study the sample complexity of program learning, we frame the problem in the *Probably Approximately Correct* (PAC) paradigm (Valiant, 1984; Vapnik and Chervonenkis, 1971; Vapnik, 1998). The goal is to learn a target function $y : \mathcal{X} \to \{\pm 1\}$ from labeled examples drawn from an unknown distribution. A learning algorithm $\mathcal{A}$ receives a sample $S$ and a hypothesis class $\mathcal{H}$ (e.g., all programs in $\mathcal{L}$ or a family of neural networks), and selects $h \in \mathcal{H}$ to minimize the generalization error $\mathrm{err}_D(h)$.

A central question in learning theory is how to design both the algorithm $\mathcal{A}$ and the hypothesis class $\mathcal{H}$ so that the number of samples $m$ required has a favorable dependence on the accuracy and confidence parameters. For the problem of program learning, we obtain the following guarantee:

**Proposition 1.** *Suppose we wish to learn a target function $y : \mathcal{X} \to \{\pm 1\}$ that can be implemented as a program of length $L$ in a programming language $\mathcal{L}$. Let $\mathcal{L}_\ell$ denote the set of programs of length $\ell$ in $\mathcal{L}$, and let $S = \{(x_i, y(x_i))\}_{i=1}^m$ be $m$ training examples drawn i.i.d. from a distribution $D$ over $\mathcal{X} \times \{\pm 1\}$. Then, with probability at least $1 - \delta$ over the draw of $S$, Alg. 2 outputs a program $h \in \mathcal{L}$ that is consistent with $S$ and satisfies $\mathrm{err}_D(h) \leq \frac{1}{m}[L \log |\Sigma| + \log(\frac{2L^2}{\delta})]$.*

This result demonstrates that if the target function can be expressed as a short program (i.e., if $L$ is small), then only a modest number of samples are required to learn it, regardless of the dimensionality of the input space. Thus, program enumeration (Alg. 2) is highly *sample efficient*. However, it remains computationally infeasible: the runtime grows exponentially in $L$. This tradeoff between sample efficiency and computational efficiency motivates our subsequent analysis.

### 2.3 GRADIENT-BASED OPTIMIZATION

To address this, deep learning replaces enumeration over the discrete program set $\mathcal{L}$ with training a neural network. We posit a parametric class $\mathcal{H} = \{h_\theta : \theta \in \Theta\} \neq \mathcal{L}$ (e.g., a neural network with learnable parameters $\theta$) and use gradient-based optimization to fit $h_\theta \in \mathcal{H}$ to data. This is typically faster than enumerating exponentially many programs, but it does not guarantee sample complexity comparable to finite-class ERM.

## 3 SGD THROUGH THE LENS OF STATISTICAL QUERIES

Next, we analyze gradient-based optimization in the *statistical query* (SQ) framework (Kearns, 1998). An SQ learner interacts with a $\tau$–tolerant oracle: given a bounded query function $\phi : \mathcal{X} \times \{\pm 1\} \to [-1, 1]$, the oracle returns $\tilde{v} = \mathbb{E}_{(x,y) \sim D}[\phi(x, y)] + \xi$, where $|\xi| \leq \tau$, with arbitrarily $\xi$ chosen.

In practice, queries are often answered by empirical averages over fresh i.i.d. batches, modeled by the 1-STAT and VSTAT oracles (Feldman et al., 2017). A 1-STAT returns $g(x, y)$ for a fresh $(x, y) \sim D$, while VSTAT($t$) returns $\mathbb{E}_D[g] \pm \tilde{O}(1/\sqrt{t})$. The two are equivalent up to polynomial overheads (Feldman et al., 2018, Thm. B.4).

The complexity of a concept class in this framework is captured by its *statistical query dimension*:

**Definition 1** (Statistical Query Dimension (Blum et al., 1994)). *For a concept class $\mathcal{C} \subseteq \{-1, +1\}^{\mathcal{X}}$ and distribution $D$ over $\mathcal{X}$, the SQ dimension $\text{SQ-DIM}_D(\mathcal{C})$ is the largest integer $d$ for which there exist $f_1, \ldots, f_d \in \mathcal{C}$ such that $\left| \mathbb{E}_{x \sim D}[f_i(x) f_j(x)] \right| \leq 1/d$ for all $i \neq j$.*

By a standard energy argument (Blum et al., 1994), if $\text{SQ-DIM}_D(\mathcal{C}) = d$, then any (stochastic) SQ learner needs $\Omega(d\epsilon^2)$ queries to reach error $\leq 1/2 - \epsilon$; see also (Reyzin, 2020, Thm. 12).

**SGD as a stochastic SQ learner.** Although stochastic gradient descent (SGD) does not query SQ oracles explicitly, each mini-batch gradient update is nothing more than the empirical average of a bounded function over $B$ fresh samples. This exactly matches the 1-STAT oracle model, with $B$ queries per iteration. Through the simulation Thm. (Feldman et al., 2018, Thm. B.4), we may therefore view $T$ iterations of mini-batch SGD with batch size $B$ as making $O(TB)$ queries to a VSTAT oracle, up to polylogarithmic factors.

Combining this observation with the SQ-dimension lower bound yields:

**Proposition 2** (Lower bound for SGD). *Let $\mathcal{C}$ be a class with $\text{SQ-DIM}_D(\mathcal{C}) = d$. Consider coordinate mini-batch SGD with batch size $B$ run for $T$ iterations. Fix $\epsilon \in (0, 1/2)$. If the algorithm outputs a hypothesis of error at most $1/2 - \epsilon$ with probability at least $2/3$, then $T \geq \Omega\left(\frac{d\,\epsilon^2}{B^{3/2}}\right)$.*

For example, for the nontrivial parity class $\mathcal{C}_{\text{par}} = \{f_s(x) = (-1)^{\langle s, x \rangle} : s \in \{0, 1\}^n\}$ under the uniform distribution on $\{0, 1\}^n$, we have $\text{SQ-DIM}_D(\mathcal{C}_{\text{par}}) = 2^n$. In particular, $T = \Omega\left((2^n \epsilon^2)/B^{3/2}\right)$. Thus, even when short program descriptions exist for high SQ-dimension classes, gradient-based learners such as SGD are inherently sample-inefficient: their query complexity grows exponentially with $n$ for parity and related tasks. Full proofs, including the formal reduction from SGD to 1-STAT to VSTAT, are deferred to App. D.1.

**Remark 1** (Sample *and* runtime complexity tradeoff). *Under the uniform distribution on $\{0, 1\}^n$, the full parity and the $k$-parity concept classes admit short programs of lengths $L = \Theta(1)$ and $L = \Theta(k \log n)$, respectively. By Prop. 1, simple program enumeration attains error $\leq \varepsilon$ with $m = \mathcal{O}(1/\varepsilon)$ samples for full parity and $m = \mathcal{O}(k \log n/\varepsilon)$ samples for $k$-parity. Its runtime is dominated by scanning all programs of length $\leq L$ over an alphabet $\Sigma$, so $\text{time(enum)} = \mathcal{O}(m |\Sigma|^L)$.*

*By contrast, in the Statistical Query (SQ) model, the SQ-dimension of full parity is $2^n$, while for $k$-parities it is $\sum_{i=0}^k \binom{n}{i} = \Theta(n^k)$ (for constant $k$). Consequently, any SQ learner with tolerance at least $1/\text{poly}(n)$—including mini-batch SGD with polynomial batch sizes—requires $2^{\Omega(n)}$ samples for full parity and $n^{\Omega(k)}$ samples for $k$-parity to reach error $< \frac{1}{2} - \gamma$. Its runtime, on the other hand, satisfies $\text{time(SGD)} = \mathcal{O}(m \times (\text{cost per gradient}) \times (\text{\# epochs}))$, which is per example/epoch $O(1)$ with respect to $L$ when the model/gradient cost does not scale with the program length $L$.*

*In short: enumeration trades low sample for time exponential in $L$, whereas SGD trades cheap per-example computation for large sample.*

## 4 METHOD

While brute-force program enumeration has good sample complexity, its runtime grows exponentially with program length, making it impractical even for modest tasks. Moreover, exhaustive search is *data-agnostic*: it enumerates programs in order of length, checking each against the data without

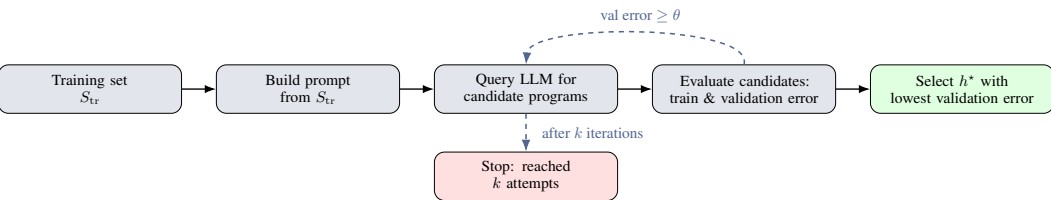

Figure 3: **An illustration of LLM-ERM.** A prompt from $S_{\mathrm{tr}}$ seeds the LLM to propose candidates, which are evaluated on train and validation sets. We track the lowest validation error and stop early when it drops below $\theta$, or otherwise after $k$ iterations.

---

**Algorithm 1** LLM-ERM: $k$-try LLM-guided search with validation

---

**Require:** $S_{\mathrm{tr}}, S_{\mathrm{val}}$; attempts $k$; prompt $\Pi$; decoding $(\tau, M)$; threshold $\theta$; optional batch $b$
**Ensure:** Program $u^\star$ with hypothesis $h^\star = [\![u^\star]\!]$ minimizing validation error
  1: Build query $\Pi(S_{\mathrm{tr}})$; initialize err$^\star \leftarrow 1$, $u^\star \leftarrow \perp$, $h^\star \leftarrow \perp$, $\mathcal{U} \leftarrow \emptyset$
  2: **for** $t = 1$ **to** $k$ **do**
  3:     Query LLM with $\Pi(S_{\mathrm{tr}})$ (temp. $\tau$, max tokens $M$) for up to $b$ candidates
  4:     **for each** $u$ in candidates with $u \notin \mathcal{U}$ **do**
  5:       $\mathcal{U} \leftarrow \mathcal{U} \cup \{u\}$; compile $u$; let $h = [\![u]\!]$
  6:       **if** $h$ undefined on some $x \in S_{\mathrm{tr}} \cup S_{\mathrm{val}}$ **then**
  7:         **continue**                                          {skip non-total/invalid candidates}
  8:       **end if**
  9:       Compute err$_{\mathrm{tr}}(h)$ and err$_{\mathrm{val}}(h)$
 10:       **if** err$_{\mathrm{val}}(h) <$ err$^\star$ **then**
 11:         (err$^\star, u^\star, h^\star) \leftarrow ($err$_{\mathrm{val}}(h), u, h)$
 12:         **if** err$^\star \leq \theta$ **then**
 13:           **return** $(u^\star, h^\star)$                              {early stop on threshold}
 14:         **end if**
 15:       **end if**
 16:     **end for**
 17: **end for**
 18: **return** $(u^\star, h^\star)$                              {best-by-validation if no early stop}

---

exploiting structure. By contrast, our approach (Alg. 1) leverages LLMs with internal reasoning (e.g., `GPT-5`), which can apply algorithmic heuristics when constructing candidates—for example, simulating the Blum–Kalai–Wasserman algorithm (Blum et al., 2003), using pattern matching, or refining proposals iteratively as performance signals develop. This adaptive search, guided by data, is the key source of efficiency of our method.

Given a labeled set of samples, we prompt the LLM to generate candidate functions (Step 3 of Alg. 1). Over $k$ iterations we collect a pool of candidates (Step 5), verify them against training and validation examples (Step 9), and select the best on the validation set with early stopping when the threshold is met (Steps 11–13). This preserves the "search-and-verify" structure of Alg. 2 but replaces exhaustive enumeration with an adaptive, LLM-guided proposal mechanism that exploits statistical cues to prioritize promising hypotheses.

**Comparison to enumeration and runtime.** Alg. 2 explores $\Omega(|\Sigma|^L)$ programs in the worst case, trading exponential *time* for near-optimal sample complexity. By contrast, LLM-ERM replaces exhaustive enumeration with an LLM-guided *propose–verify* loop: the search is effectively restricted to at most $k\,b$ candidates, each selected using data-informed heuristics (e.g., parity checks, divisibility filters) surfaced by the LLM. While this does not provide worst-case guarantees, the ability of modern "thinking" LLMs to simulate algorithmic strategies, exploit residuals, and adapt proposals yields a practical trade-off: dramatically narrower search, with ERM-style selection preserving the generalization benefits of learning short programs. For runtime, fix $k$, $b$, and $m$, and let $T_{\mathrm{LLM}}$ and $T_{\mathrm{ver}}$ be the average time for one LLM call and the per-example verification. Each iteration performs one LLM call and verifies up to $b$ candidates on $m$ examples, so the per-iteration cost is $T_{\mathrm{LLM}} + b\,m\,T_{\mathrm{ver}}$, and the total time is $\mathrm{Time}(k, b, m) = \mathcal{O}\left(k\,T_{\mathrm{LLM}} + k\,b\,m\,T_{\mathrm{ver}}\right)$. For small $m$ the

---

**LLM Prompt**

**Problem Statement:** Given a sequence of input vectors (binary, length {sequence_dimension}) and their corresponding scalar binary outputs ('0' or '1'), find a concise Python function `f(x)` that accurately approximates the underlying relationship. The function should not be a trainable model, but a direct logical or mathematical representation of the target function.
**Data Examples:**

```
00011110101111001010010100110 -> 1
... 0110110101110000100101010001000 -> 1
```

**You must output ONLY a single JSON object: {"code": "<python function>"}**

---

Figure 4: Prompt used in our LLM-ERM procedure. We run `GPT-5` with this prompt for up to $k$ independent iterations, each returning only Python code for a candidate target function.

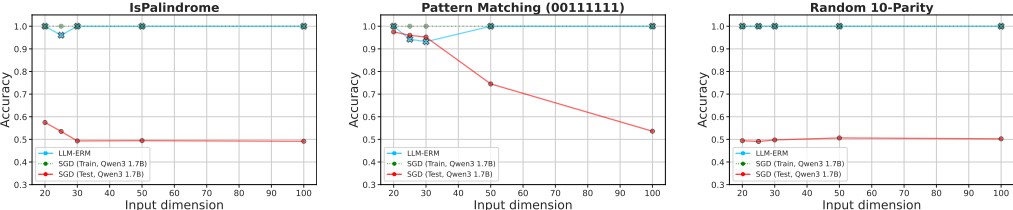

Figure 5: **LLM-ERM generalizes from 200 samples, while SGD-trained LLM overfits.** With only 200 training examples per task, LLM-ERM typically recovers the target function exactly, whereas SGD training of `Qwen3-1.7B` from scratch fits the training data but fails to generalize on most tasks when $n$ is sufficiently large. See Fig. 7 for additional results.

LLM call dominates; for large $m$ verification dominates. Thus, wall clock time scales linearly in $k$, $b$, and $m$, with constants set by LLM efficiency and verification cost.

**Interpretability.** LLM-ERM makes both the *learned object* and the *learning process* transparent. Each run returns (i) an *executable program* and (ii) a *reasoning trace* recording candidates, rationales, and residual errors (with train accuracies when the model chose to compute them). Fig. 1 shows a `GPT-5-Thinking` reasoning trace (taken from the ChatGPT-5 web UI) on IsPrime and Random 10-Parity with 100 samples of 50-digit inputs (see also Fig. 15). The trace ties concrete rules (e.g., parity checks, digit filters, Miller–Rabin) to the errors they address, clarifying *why* a candidate is proposed, *which* mistakes it fixes, and *when* search stops. In particular, failure modes become auditable (e.g., partial solutions on palindromes or Dyck-2), and successes are inspectable, with invariants testable via counterfactual probes. Because the output is symbolic code, we can unit test, stress test out of distribution, or edit and re-run components—turning behavior into an executable, inspectable artifact rather than opaque weights. The trace thus serves as a compact, reproducible *proof of learning*, documenting not only the final program but also the path to it.

## 5 EXPERIMENTS

We evaluate LLM-ERM's sample efficiency against SGD-trained neural networks on synthetic algorithmic tasks under controlled distributions (e.g., parity, pattern matching, palindromes, Dyck-2, primality, cellular automata; see App. C.1). For each sequence length we use a small-data regime ($m = 200$ labeled examples; 100 train and 100 validation) and test on large held-out sets. LLM-ERM proposes $k$ candidate programs from a pretrained reasoning LLM and selects the one with the lowest validation error, while baselines are transformers trained from scratch with SGD on the same data. Unless noted otherwise, hyperparameters and preprocessing are held fixed across tasks.

**Training details.** We train LLMs from scratch as binary classifiers $h$ for targets $y : \mathcal{X} \to \{0, 1\}$. We draw $m$ i.i.d. samples $S = \{(x_i, y(x_i))\}_{i=1}^m$ with $\mathcal{X} = \{0, 1\}^n$ or $\mathcal{X} = \{0, \dots, 9\}^n$. Each pair $(x_i, y(x_i))$ is represented as a sequence of length $n+1$: the model reads $x_i = (x_{i,1}, \dots, x_{i,n})$ and predicts $y(x_i)$. We optimize binary cross-entropy between the model's logits and the ground-truth labels, using AdamW for 200 epochs with cosine annealing ($\eta_{\max}=10^{-5}$, $\eta_{\min}=10^{-6}$).

| Task | $n=20$ | | $n=25$ | | $n=30$ | | $n=50$ | | $n=100$ | |
|------|---------|---------|---------|---------|---------|---------|---------|---------|---------|---------|
| | Baseline | LLM-ERM | Baseline | LLM-ERM | Baseline | LLM-ERM | Baseline | LLM-ERM | Baseline | LLM-ERM |
| Full Parity | 50.5% | $\infty$% | 50.1% | $\infty$% | 50.1% | $\infty$% | 50.0% | $\infty$% | 49.3% | $\infty$% |
| First-Half Parity | 51.0% | **100%** | 51.3% | **100%** | 48.9% | **100%** | 50.5% | **100%** | 50.6% | **100%** |
| Random 3-Parity | 49.8% | **100%** | 50.5% | **100%** | 50.4% | **100%** | 49.9% | **100%** | 49.7% | **100%** |
| Random 10-Parity | 49.4% | **100%** | 49.1% | **100%** | 49.8% | **100%** | 50.6% | **100%** | 50.3% | **100%** |
| Pattern Matching (10101010) | 91.4% | $\infty$% | 82.8% | **98.9%** | 57.8% | **98.5%** | 58.7% | $\infty$% | 51.5% | $\infty$% |
| Pattern Matching (00111111) | 97.5% | $\infty$% | 96.0% | 94.2% | 95.2% | 93.2% | 74.5% | $\infty$% | 53.6% | $\infty$% |
| IsPalindrome | 57.5% | **100%** | 53.5% | **96.0%** | 49.3% | **100%** | 49.4% | **100%** | 49.2% | **100%** |
| Dyck-2* | 59.8% | **77.4%** | 58.0% | **90.5%** | 53.0% | **80.0%** | 51.1% | **90.5%** | 51.4% | **80.1%** |
| IsPrime | 88.5% | $\infty$% | 88.5% | $\infty$% | 87.8% | $\infty$% | 89.9% | $\infty$% | 90.3% | $\infty$% |
| IsPrime (Ends in {1, 3, 7, 9}) | 59.9% | $\infty$% | 60.2% | $\infty$% | 57.0% | $\infty$% | 57.0% | $\infty$% | 58.8% | $\infty$% |
| Cellular Automata Parity$^\diamond$ | 49.4% | **100%** | 50.1% | **100%** | 49.8% | $\infty$% | 50.5% | $\infty$% | 49.4% | $\infty$% |
| SHA-256 Parity | 48.3% | **50.2%** | **50.2%** | 49.9% | **50.5%** | 50.4% | **50.3%** | 50.0% | 49.8% | **50.1%** |

*For Dyck-2, lengths are $n=\{20, 40, 60, 80, 100\}$ respectively.
$^\diamond$For Cellular Automata Parity, length $n=\{100\}$ took $k=27$ attempts.

Table 1: **Test accuracy of SGD-trained LLMs vs. LLM-ERM.** The baseline model attains 100% training accuracy on all tasks but fails to generalize, often collapsing to chance-level performance ($\approx 50$%). By contrast, LLM-ERM achieves near-perfect generalization by synthesizing functionally correct programs. In some cases, LLM-ERM produces dimension-invariant Python programs; in these cases, test accuracy is denoted as $\infty$%.

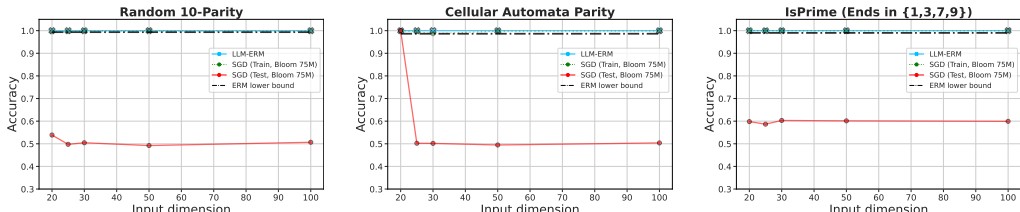

Figure 6: **SGD-trained LLMs struggle on algorithmic tasks even with 100k samples.** We train `Bloom-75M` on Random 10-Parity **(left)**, Cellular Automata Parity **(middle)**, and IsPrime with negatives restricted to $\{1, 3, 7, 9\}$ **(right)**, each with 100k examples. Despite abundant data and perfect fitting, the model still overfits and fails to generalize. The ERM lower bound is computed as $\frac{1}{m}[L \log |\Sigma| + \log(\frac{2L^2}{\delta})]$ with $\delta=10^{-10}$, $|\Sigma|=128$ (ASCII), and $L$ the length of a short Python program implementing the target.

We constructed an experiment using a custom-configured model based on Bloom architecture (Workshop et al., 2023). We scaled down the `Bloom-560M` configuration to a 75.7M parameter model by reducing the hidden dimension to 512 and the number of attention heads to 8, while keeping the number of layers at 24 and preserving the head-to-dimension ratio. This smaller model was trained from scratch on a significantly larger dataset of $m_{\text{train}}=100$k samples for an extended 1000 epochs. We used a larger batch size of 256 and a constant $\eta=10^{-5}$. We train on sequence lengths $n \in \{20, 25, 30, 50, 100\}$, with $m_{\text{train}}=200$ training examples (for all other models) and a held-out test set of $m_{\text{test}}=10,000$ examples. Batch size is 20. All runs use `bfloat16` on a single node with two 94 GB NVIDIA H100 GPUs.

**Model architecture.** For the baseline, we train a Qwen3-1.7B model (Yang et al., 2025) from scratch, adapted to binary classification: the vocabulary is restricted to three tokens (`vocab_size=3`) and the language-modeling head is replaced by a single linear layer (`hidden_size→1`). The network has 28 transformer layers, 16 attention heads, and hidden size 2048 (about 1.4B parameters). For additional experiments with Llama 3.2 1B (Meta AI, 2024; Grattafiori et al., 2024) and DeepSeek-Coder 1.3B (Guo et al., 2024) see Fig. 8 (in App. C).

**Evaluation tasks.** We use several binary algorithmic problems spanning (i) local pattern detection, (ii) global XOR–style dependencies (parity function variants), (iii) symmetry/mirroring (palindrome detection), (iv) context-free parsing (Dyck-2), (v) and number-theoretic predicates (primality). For each sequence length, datasets are class-balanced with equal positives and negatives (see App. C.1 for formal definitions and data-generation procedures). In essence, each task presents a different challenge for learning complex reasoning patterns. We systematically test generalization across tasks and analyze performance over varying input lengths and training durations.

**LLM-ERM.** Our method (Alg. 1, illustrated in 3) leverages in-context program synthesis with a pretrained LLM (GPT-5). We split $S$ into equal-sized training ($m_{\text{train}}{=}100$) and validation sets ($m_{\text{validation}}{=}100$). The training split conditions the prompt in Fig. 4, which we submit to the LLM as shown in Fig. 3. The generation process is configured with `reasoning_effort = High`, max tokens 20k, and a per call timeout 20 mins. Temperature and top-p are managed by the platform and are not user-configurable. We generate up to $k{=}5$ candidate responses with batch sizes $b{=}1$ and evaluate each on the validation set. We set $\theta{=}0$ and stopped once validation error reached zero. If no perfect program appeared, we kept the candidate with the lowest validation error overall.

RESULTS

We observe a sharp performance split between the SGD baseline and LLM-ERM. SGD often reaches perfect training accuracy yet fails to generalize. In contrast, LLM-ERM reliably recovers the underlying rule from only a few examples and generalizes well. We summarize the results in Tab. 1.

**SGD fails to generalize.** The baseline, a Qwen3 1.7B model trained from scratch, exhibited obvious signs of severe overfitting. Across all tasks and sequence lengths, the model achieved 100% training accuracy, indicating sufficient capacity to memorize the 200 training examples. However, its performance on the held-out test set of 10,000 samples generally shows a near-total failure to learn the underlying algorithmic principles.

**Failure on non-local and recursive tasks.** For tasks requiring non-local reasoning or recursive structures, the model's performance was statistically indistinguishable from random guessing. On all variants of parity (Full, First-Half, Random 3- and 10-Parity), Palindrome Recognition, and Double Parentheses (Dyck-2 language), the test accuracy hovered around 50% (see Tab. 1 and Fig. 7). This indicates a catastrophic failure to generalize. The model did not capture the global property of parity, the symmetrical structure of palindromes, or the context-free grammar of the Dyck language, instead relying on memorization of the training data.

**SGD has limited success on local/heuristic tasks.** The model succeeds mainly when simple local cues suffice. In Pattern Matching, accuracy is high at small $n$ but collapses as $n$ grows: for `00111111`, 97.5% at $n{=}20$ vs. 53.6% at $n{=}100$; for `10101010`, 91.4% at $n{=}20$ vs. 51.5% at $n{=}100$. This indicates the model likely learned a brittle local detector rather than a robust search procedure. For IsPrime, $\approx 90\%$ test accuracy largely reflects a last-digit heuristic (last digits $0, 2, 4, 5, 6, 8$ imply non-prime), which alone yields $\approx 80\%$ on a balanced distribution. When we restrict all negatives to end in $\{1, 3, 7, 9\}$, accuracy drops to $\approx 60\%$ (a pure last-digit rule would be $\approx 50\%$), confirming heavy reliance on the final digit.

**Achieving perfect algorithmic discovery via LLM-ERM.** For the most complex tasks, LLM-ERM succeeded where SGD failed completely. On all Parity variants, Palindrome Recognition, and Primality, the method consistently generated a functionally correct Python program, achieving *100% test accuracy across all sequence lengths*. These results demonstrate the ability of LLM-guided synthesis to move beyond statistical correlation and perform genuine algorithmic induction. In Tab. 1 we numerically compare the two methods, where our method achieves perfect algorithmic discovery on several tasks, including parity variants and primality testing (we denote that by $\infty\%$).

**Scaling to 100k samples does not fix generalization.** With enough training data, SGD is expected to generalize well, assuming it can fit the training set. While LLM-ERM learns these tasks (except SHA-256 Parity) from just 200 samples, we ask whether SGD can match this performance even with $500\times$ more data. Fig. 6 Tab. 6 shows that an SGD-trained `Bloom` still overfits: Random 10-Parity and Cellular Automata Parity remain near chance, and IsPrime stays weak when negative samples are constrained to end in a digit from $\{1, 3, 7, 9\}$. Thus, even with 100k samples, SGD fails to generalize. The ERM lower bound, based on Prop. 1, is computed as $1 - \frac{1}{m}[L \log |\Sigma| + \log(\frac{2L^2}{\delta})]$ with $m{=}100k$, $\delta{=}10^{-10}$, $|\Sigma|{=}128$ (ASCII), and $L$ the length of a short Python program for the target function. This bound guarantees that Alg. 2 performs near $100\%$ at test time when $m = 100k$. However, SGD does not generalize, whereas LLM-ERM learns these tasks perfectly with only 200 samples.

**Ablations.** We performed ablations with various architecture from scratch (App. C.2.1), fine-tuning (App. C.2.1), in-context learning ( C.2.1), learning rates (App. C.2.3) and batch sizes (App. C.2.2). Across the different settings, results are consistent: SGD fails to generalize on sparse, non-local tasks and learns only length-dependent heuristics on Pattern Matching and IsPalindrome (App. C.2).

# 6 REPRODUCIBILITY STATEMENT

We took care to make all experiments transparent and repeatable. The main text, figures, and appendix specify the exact data generators and preprocessing rules for each task, together with the train/validation/test splits and random seeds; all scripts that regenerate the datasets and splits are included in the (anonymous) code release, along with pinned package versions and hardware notes (GPU type and precision). For LLM-ERM we provide the full prompt template (Fig. 4), decoding settings (reasoning effort, text verbosity, max tokens), $(k, b)$, early-stopping threshold $\theta$, and timeout values, plus the raw candidate programs and validation logs used to select $h^\star$ (Fig. 3, Alg. 1). For SGD baselines we include model configs, optimization hyperparameters, learning-rate schedules, batch sizes, number of epochs, and evaluation protocols. Because LLM-ERM queries a hosted `GPT-5` API, provider-side updates may introduce small run-to-run variation; to mitigate this we fix seeds and decoding settings, and report the results against a wide range of tasks.

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

## A  LIMITATIONS

Although LLM-ERM achieves strong results across diverse tasks, several caveats remain. In *Pattern Matching*, solutions are usually correct but sometimes include small implementation errors (e.g., 93.2% accuracy for `00111111` at $n=30$), highlighting sensitivity to prompt phrasing and decoding choices. *Dyck-2* proves more challenging (77.4–90.5% test accuracy), underscoring the difficulty of reliably synthesizing parsers for context-free structures. On cryptographic-style problems such as `SHA-256` Parity, the model achieves only chance-level accuracy, which is due to the pseudorandom nature of the target function.

More broadly, our evaluation focuses on realizable, discrete algorithmic problems with inexpensive verifiers. Extending LLM-ERM to more complex settings—such as tasks with noisy labels, approximate or continuous objectives, or costly/long-running executors—remains an open direction. Performance also depends on factors beyond our control, including pretrained model priors, prompt design, decoding parameters, and the candidate budget $(k, b)$. These sensitivities suggest opportunities for principled prompt optimization, adaptive decoding strategies, and integration with stronger verification pipelines in future work.

## B  LLM USAGE STATEMENT

**Models and access.**  We used a hosted `GPT-5` API and `GPT-5-thinking` (ChatGPT-5 web UI) during 9/1/2025-9/24/2025. At this time, temperature and top-p cannot be modified; these settings are controlled by the platform. Decoding settings `max_output_tokens=20k`, `reasoning_effort=High`, `text_verbosity=Low` are used.

**Role in the research workflow.**  (1) *Method (LLM-ERM).* The LLM served as a proposal generator for candidate Python programs conditioned on training examples; selection was performed automatically via ERM on a held-out validation set. (2) *Writing and proofing.* LLMs were used for copy-editing, clarity edits, and LaTeX refactors. They were also used to proofread and refine proofs (e.g., tightening inequalities, suggesting alternative lemma structures). All formal statements and proofs in the paper were authored, verified, and, where needed, re-derived by the authors. (3) *Code.* LLMs were used to generate small code snippets within our codebase (e.g., data preprocessing utilities, hyperparameter sweeps, test harness helpers, and non-critical boilerplate). All such snippets were reviewed, modified as needed, and validated by the authors with unit tests and static checks before use. The core experiment logic (dataset generators, verifiers, and evaluation scripts) was authored and audited by the authors. (4) *Ideation and experimental design.* Research questions, task definitions, and experimental protocols were conceived by the authors; LLMs were not used to originate these.

**Verification and reproducibility.**  All LLM-generated artifacts (text and code) were checked by the authors. Candidate programs produced by the `GPT-5` were evaluated deterministically on fixed precomputed data splits. Because we depend on a hosted API, provider-side updates may introduce small run-to-run variation; we mitigate this by fixing decoding settings where possible and releasing exact prompts/settings.

**Models used as experimental subjects (SGD baselines).**  In addition to using `GPT-5` as a core component of our LLM-ERM method, we trained open-source LLMs (e.g., `Qwen3-1.7B`, `DeepSeek-Coder-1.3B`, `Llama 3.2-1B`) from scratch as SGD baselines. These models were *not* used as assistants for writing, proof checking, or code suggestion; rather, they served solely as architectures to evaluate SGD-based training.

**Authorship.**  No LLM is listed as an author. The authors take full responsibility for the paper's content. We disclose LLM usage here in accordance with the policy.

## C  ADDITIONAL EXPERIMENTAL DETAILS AND RESULTS

### C.1  EVALUATION TASKS

We evaluate both methods across a diverse suite of algorithmic tasks designed to probe different facets of logical reasoning, from simple pattern recognition to complex, non-local computations. For

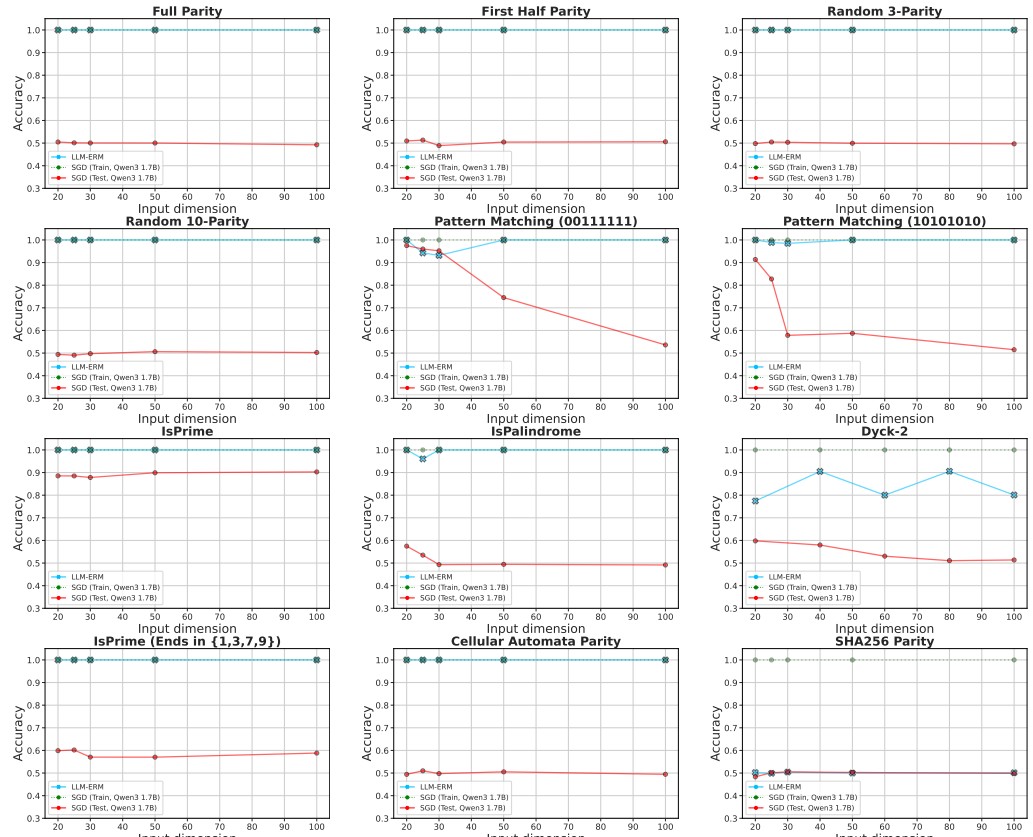

Figure 7: **LLM-ERM generalizes from 200 samples, while SGD-trained LLM overfits.** With only 200 training examples per task, LLM-ERM typically recovers the target function exactly, whereas SGD training of Qwen3-1.7B from scratch fits the training data but fails to generalize on most tasks when $n$ is sufficiently large. Due to the extreme pseudo-random behavior of the SHA-256 function, it remains difficult to learn by both LLM-ERM and SGD.

each task, the training and test datasets are balanced with an equal number of positive and negative examples. (Add a footnote, for dimension 20 in Dyck-2 it is impossible to create sufficient positive labels, hence test set here is reduced to 1000 samples).

- **Parity.** Parity functions are functions of the form $(-1)^{\langle s,x \rangle}$, where $s \in \{0,1\}^n$ is a fixed binary vector. We experiment with multiple types of parity functions: the full parity function $(-1)^{\langle \mathbf{1}_n, x \rangle}$ (where $\mathbf{1}_n = (1,\ldots,1)$ of length $n$), the first-half parity function $(-1)^{\langle (\mathbf{1}_{n/2} \| \mathbf{0}_{n/2}), x \rangle}$ (where $(\mathbf{1}_{n/2} \| \mathbf{0}_{n/2})$ is the concatenation of $\mathbf{1}_{n/2} = (1,\ldots,1)$ and $\mathbf{0}_{n/2} = (0,\ldots,0)$), random $k$-parity, which is a function of the form $(-1)^{\langle s,x \rangle}$ with a random vector $s$ with $k$ 1s and $n - k$ zeros.

- **Pattern Matching.** For a fixed pattern $p \in \{0,1\}^k$ with $k < n$, the label is $y(x) = \mathbb{I}\big[\exists i \in \{1,\ldots,n-k+1\}$ such that $(x_i,\ldots,x_{i+k-1}) = p\big]$, where $\mathbb{I}[\cdot]$ is the indicator. We use patterns 10101010 and 00111111 to assess local feature detection.

- **IsPalindrome.** This function is defined as $y(x) = \mathbb{I}\big[\forall i \in \{1,\ldots,\lfloor n/2 \rfloor\} : x_i = x_{n-i+1}\big]$. Positive examples (palindromes) are constructed by mirroring a random first half. For negatives, we generate a palindrome and flip a single bit in the first half, thereby testing sensitivity to precise symmetric structure.

- **Dyck-2.** Let $\mathcal{M} : \{0,1\}^2 \to \{`(`,`)`,`[`,`]`\}$ be a mapping from bit-pairs to characters, and let $S(x)$ be the resulting character string. The function is $y(x) = \mathbb{I}[S(x) \in D_2]$, where $D_2$ is the Dyck-2 formal language. This task assesses the ability to recognize a context-free language, which requires stack-like, recursive reasoning.

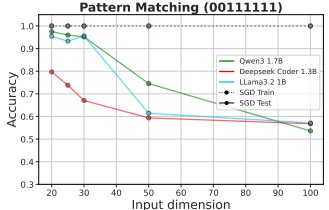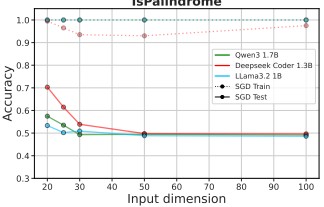

Figure 8: **Different LLM architectures consistently overfit the training data.** We compare architectures on three tasks—Random 10-Parity, Pattern Matching (00111111), and IsPalindrome. All models exhibit the same issue described above: they fit the training data but fail to generalize when the sequence length is too large.

| Task | Qwen3 1.7B | | | | | Deepseek-Coder 1.3B | | | | | Llama3.2 1B | | | | |
|---|---|---|---|---|---|---|---|---|---|---|---|---|---|---|---|
| | $n=20$ | $n=25$ | $n=30$ | $n=50$ | $n=100$ | $n=20$ | $n=25$ | $n=30$ | $n=50$ | $n=100$ | $n=20$ | $n=25$ | $n=30$ | $n=50$ | $n=100$ |
| **Rand. 10-Parity** | 49.4% | 49.1% | 49.8% | 50.6% | 50.3% | 49.7% | 49.5% | 50.3% | 49.0% | 50.2% | 50.6% | 50.2% | 49.4% | 49.5% | 49.7% |
| **Pattern Matching (00111111)** | 97.5% | 96.0% | 95.2% | 74.5% | 53.6% | 79.7% | 73.8% | 67.0% | 59.4% | 56.8% | 95.4% | 93.3% | 95.5% | 61.4% | 57.1% |
| **IsPalindrome** | 57.5% | 53.5% | 49.3% | 49.4% | 49.2% | 70.3% | 61.5% | 53.9% | 49.8% | 49.7% | 53.3% | 50.2% | 50.9% | 48.8% | 48.6% |

Table 2: **Test accuracy rates across different model architectures.** All models demonstrate failure on the Random 10-Parity task. For IsPalindrome and Pattern Matching, accuracy is high for shorter sequences but degrades significantly as sequence length increases.

- **IsPrime.** The input sequence $x = (x_1, \ldots, x_n)$ encodes a base-10 integer with digits $x_i \in \{0, \ldots, 9\}$. The label is $y(x) = \mathbb{I}[\text{IsPrime}(\text{int}(x))]$. This task requires arithmetic and number-theoretic reasoning, posing a challenge for neural networks. The dataset comprises equal numbers of randomly sampled $n$-digit primes and $n$-digit non-primes, each drawn uniformly from its respective set.

- **IsPrime (Ends in {1,3,7,9}).** The function is unchanged, $y(x) = \mathbb{I}[\text{IsPrime}(\text{int}(x))]$, but the dataset is constrained so that the last digit $x_n \in \{1, 3, 7, 9\}$. This removes the most common statistical shortcuts for primality and forces reliance on number-theoretic properties that depend on the entire sequence.

- **Cellular Automaton Parity.** The label is $y(x) = \left( \sum_{i=1}^{n} x_i' \right) \bmod 2$, where $x' = (x_1', \ldots, x_n')$ is derived from $x$ by a local update. Each bit $x_i'$ depends on its neighborhood $(x_{i-1}, x_i, x_{i+1})$ via $x_i' = x_{i-1} \oplus (x_i \lor x_{i+1})$. We use boundary conditions $x_0 = x_{n+1} = 0$. The task combines a local, nonlinear (and potentially chaotic) transform with a global parity computation.

- **SHA-256 Parity.** Let $(h_1, \ldots, h_{256}) = \text{SHA-256}(x)$ be the 256-bit hash of $x$. The label is $y(x) = \left( \sum_{i=1}^{256} h_i \right) \bmod 2$. Because cryptographic hashes are effectively pseudorandom, this task is a stringent test of a model's ability to learn highly complex, nonlinear dependencies.

## C.2 COMPARISONS BETWEEN LLM-ERM AND SGD

To test the robustness of our main result—that SGD fails to achieve algorithmic generalization—we ran ablations reported in the appendix: (i) fine-tuning pretrained LLMs in place of training from scratch (App. C.2.1), (ii) model architecture (App. C.2.1, Tab. 2), (iii) learning rate (App. C.2.3), and (iv) batch size (App. C.2.2). Across all settings, models fit the training data yet failed to achieve algorithmic generalization; this pattern persists over wide hyperparameter sweeps and with fine-tuning, indicating the effect is not an artifact of configuration choices but a limitation of SGD on these tasks.

### C.2.1 TESTING DIFFERENT MODELS

**Training from scratch.** To determine if the poor generalization was specific to the Qwen3 1.7B architecture, we replicated our SGD experiments with two other prominent open-source models of similar scale: Deepseek-Coder-1.3B and Llama3.2-1B. Each model was trained from scratch under identical conditions, using the same hyperparameters and training data as described in our main experiments. We evaluated them on a representative subset of tasks: IsPalindrome (non-

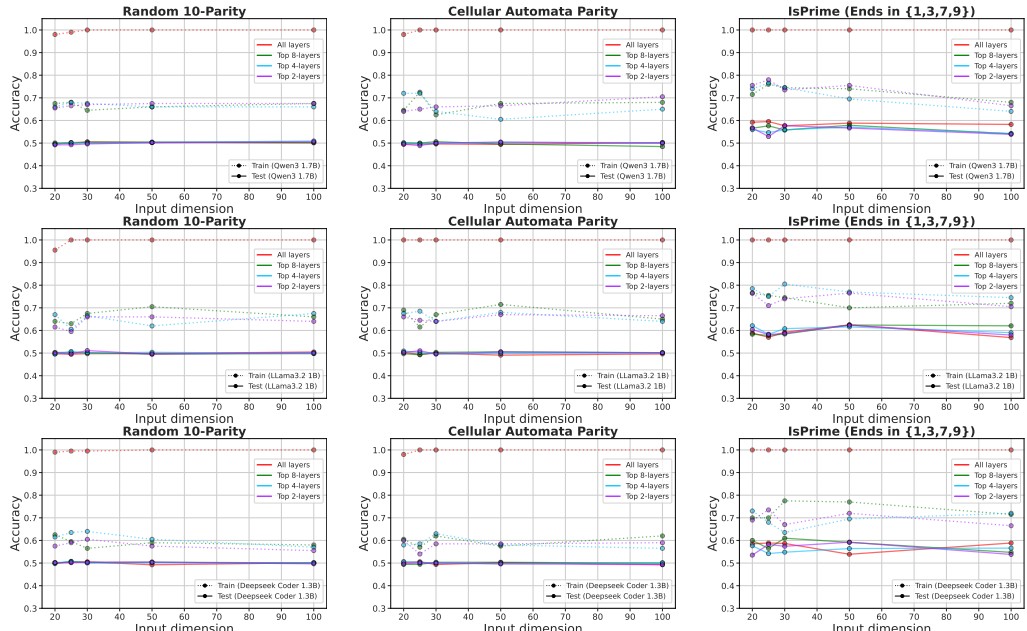

Figure 9: **Fine-tuning pre-trained LLMs fails to overcome overfitting on algorithmic tasks.** We fine-tuned `Qwen3 1.7B`, `Llama3.2 1B`, and `Deepseek Coder 1.3B` on three tasks with 200 samples, training either the full model or only the top 2, 4, or 8 layers. While models could partially fit the data—or perfectly fit it when the full network was fine-tuned—their test accuracy remained near 50%.

local reasoning), Random 10-Parity (sparse non-local reasoning), and Pattern Matching (00111111) (heuristic-based reasoning).

As shown in Tab. 2 and Fig. 8, the choice of architecture had no significant impact on generalization performance. For Random 10-Parity, all three models: Qwen3, Deepseek-Coder, and Llama3.2, performed at chance level on the test set (50% accuracy), confirming a consistent failure to learn sparse non-local dependencies. For IsPalindrome and Pattern Matching, the models achieved high accuracy on shorter sequences but failed to generalize as sequence length increased, with performance degrading significantly towards chance level. This consistency across architectures strongly suggests that the failure to generalize is a fundamental limitation of the SGD paradigm for these complex algorithmic tasks, where models learn shallow heuristics that do not scale with problem size, rather than a deficiency of a particular model.

**Fine-tuning pre-trained models.** A natural objection to our results is that LLM-ERM may succeed only because it leverages large-scale pretraining before exposure to algorithmic problems. To test this, we fine-tuned three pre-trained LLMs—`Qwen3 1.7B` Yang et al. (2025), `Llama3.2 1B` (Meta AI, 2024), and `Deepseek Coder 1.3B` Guo et al. (2024)—on Random 10-Parity, Cellular Automata Parity, and IsPrime (restricted to numbers ending in $1, 3, 7, 9$). Fine-tuning used 200 training samples and 10k test samples, with input lengths $n \in 20, 25, 30, 50, 100$. Models were trained for 1000 epochs with AdamW and CosineAnnealingLR (Loshchilov and Hutter, 2019) (batch size 20, `bfloat16`), using space-separated integer tokenization (EOS padding) and a single-logit LM head (hidden$\rightarrow$1) at the final position.

We evaluated four regimes Tab. 3 and Fig. 9: fine-tuning the whole model and partial fine-tuning of only the top 2, 4, or 8 transformer blocks. Full fine-tuning typically achieved (near-)perfect training accuracy, while partial fine-tuning produced moderate fits. However, test performance remained essentially unchanged: on tasks requiring non-local dependencies (Random 10-Parity, Cellular Automata Parity), accuracy stayed at chance ($\approx 50\%$) across models, lengths, and depths. On IsPrime, fine-tuning yielded only modest improvements (e.g., $62.5\%$ at $n{=}50$ for Llama3.2 full FT). Overall, no model under any fine-tuning regime demonstrated generalization on either of these tasks.

| | | Fine-Tuning Pre-trained Models (Test Accuracy) | | | | | | | | | | | | | |
| | | Llama3.2 1B | | | | | Qwen3 1.7B | | | | | Deepseek-Coder 1.3B | | | | |
| Task | Layers Tuned | $n=20$ | $n=25$ | $n=30$ | $n=50$ | $n=100$ | $n=20$ | $n=25$ | $n=30$ | $n=50$ | $n=100$ | $n=20$ | $n=25$ | $n=30$ | $n=50$ | $n=100$ |
|---|---|---|---|---|---|---|---|---|---|---|---|---|---|---|---|---|
| **Random 10-Parity** | Top 2 | 50.1% | 49.6% | 51.1% | 49.5% | 50.2% | 49.2% | 49.3% | 49.6% | 50.2% | 50.6% | 50.2% | 50.2% | 50.4% | 50.4% | 49.7% |
| | Top 4 | 49.5% | 50.7% | 50.0% | 50.3% | 49.8% | 49.5% | 50.3% | 49.9% | 50.3% | 50.9% | 50.2% | 50.7% | 50.0% | 50.2% | 50.1% |
| | Top 8 | 50.3% | 50.3% | 50.2% | 49.5% | 49.8% | 50.1% | 49.9% | 50.5% | 50.5% | 50.1% | 49.7% | 50.8% | 50.2% | 50.4% | 50.1% |
| | Full Model | 49.7% | 49.5% | 49.7% | 49.8% | 50.5% | 49.7% | 50.2% | 50.6% | 50.2% | 50.2% | 50.1% | 50.3% | 50.5% | 49.2% | 50.1% |
| **IsPrime (Restricted)** | Top 2 | 60.4% | 58.2% | 58.7% | 62.3% | 58.0% | 56.7% | 52.9% | 57.7% | 56.7% | 53.9% | 53.5% | 58.2% | 57.4% | 59.2% | 53.8% |
| | Top 4 | 62.1% | 58.3% | 60.8% | 61.6% | 59.0% | 56.0% | 54.7% | 55.9% | 57.1% | 54.2% | 57.6% | 54.2% | 54.8% | 56.4% | 56.6% |
| | Top 8 | 58.3% | 57.8% | 58.4% | 62.4% | 62.1% | 56.6% | 57.6% | 55.8% | 57.9% | 54.1% | 60.0% | 56.5% | 61.0% | 59.2% | 54.7% |
| | Full Model | 58.9% | 57.0% | 59.2% | 62.5% | 56.9% | 59.2% | 59.5% | 57.7% | 58.8% | 58.3% | 58.5% | 58.9% | 58.7% | 53.9% | 58.9% |
| **Cellular Automata Parity** | Top 2 | 50.3% | 51.0% | 49.6% | 50.5% | 50.1% | 49.5% | 48.9% | 50.1% | 50.4% | 49.9% | 49.6% | 50.1% | 50.0% | 49.7% | 49.3% |
| | Top 4 | 50.9% | 50.2% | 49.7% | 49.8% | 50.2% | 50.1% | 49.7% | 49.9% | 50.3% | 50.2% | 50.7% | 50.3% | 50.1% | 49.7% | 50.2% |
| | Top 8 | 49.7% | 49.2% | 50.3% | 50.5% | 50.2% | 50.1% | 50.0% | 50.6% | 49.6% | 48.5% | 49.4% | 49.4% | 50.3% | 50.3% | 50.0% |
| | Full Model | 50.3% | 49.4% | 50.0% | 49.1% | 49.6% | 49.5% | 49.7% | 49.6% | 49.4% | 50.2% | 50.2% | 50.6% | 49.3% | 50.2% | 49.3% |

| | In-Context Learning (Test Accuracy) | | | | | | | | | | | | | | |
| Task | Qwen3-30B-Instruct | | | | | Qwen3-Coder-30B-Instruct | | | | | Deepseek-Coder-33B-Instruct | | | | |
| | $n=20$ | $n=25$ | $n=30$ | $n=50$ | $n=100$ | $n=20$ | $n=25$ | $n=30$ | $n=50$ | $n=100$ | $n=20$ | $n=25$ | $n=30$ | $n=50$ | $n=100$ |
|---|---|---|---|---|---|---|---|---|---|---|---|---|---|---|---|
| **Full Parity** | 52.0% | 43.0% | 51.0% | 38.0% | 53.0% | 49.0% | 47.0% | 47.0% | 50.0% | 50.0% | 51.0% | 47.0% | 54.0% | 51.0% | 43.0% |
| **Random 10-Parity** | 45.0% | 53.0% | 54.0% | 51.0% | 50.0% | 50.0% | 49.0% | 53.0% | 42.0% | 44.0% | 51.0% | 55.0% | 55.0% | 35.0% | 48.0% |
| **IsPalindrome** | 58.0% | 48.0% | 49.0% | 49.0% | 47.0% | 52.0% | 53.0% | 51.0% | 52.0% | 51.0% | 56.0% | 47.0% | 63.0% | 47.0% | 51.0% |
| **Cellular Automata Parity** | 47.0% | 45.0% | 44.0% | 43.0% | 49.0% | 50.0% | 47.0% | 44.0% | 49.0% | 49.0% | 52.0% | 42.0% | 46.0% | 50.0% | 44.0% |
| **IsPrime (Restricted)** | 53.0% | 53.0% | 53.0% | 57.0% | 50.0% | 52.0% | 49.0% | 50.0% | 50.0% | 47.0% | 51.0% | 49.0% | 47.0% | 53.0% | 53.0% |
| **IsPrime** | 57.0% | 62.0% | 56.0% | 66.0% | 52.0% | 59.0% | 55.0% | 53.0% | 51.0% | 49.0% | 47.0% | 57.0% | 45.0% | 47.0% | 57.0% |

Table 3: **Test accuracy for fine-tuning and in-context learning fails to generalize on algorithmic tasks. (Top)** Fine-tuning pre-trained models ( 1B scale) on 200 examples fails on non-local tasks (chance accuracy) and yields only marginal gains on heuristic-based ones, regardless of the number of layers tuned. **(Bottom)** In-context learning with larger instruction-tuned models (30B+ scale) and the same 200 examples also fails to generalize.

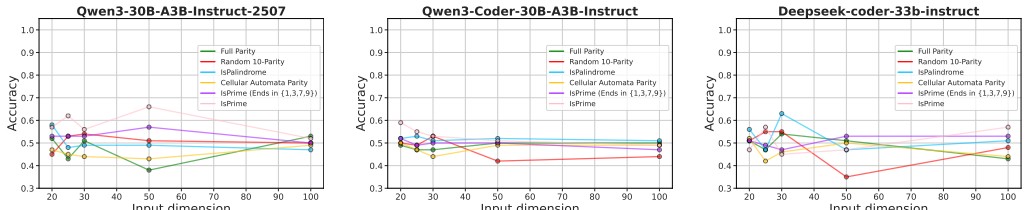

Figure 10: **Test accuracy for in-context learning with three large models also fails to generalize.** The plots display the performance of `Qwen3-30B-A3B-Instruct -2507`(left), `Qwen3-Coder-30B -A3B-Instruct`(middle), and `Deepseek-Coder-33B -Instruct`(right) on classification tasks when provided with 200 training examples in-context. Results are shown for six different tasks. For all three models, the observed test accuracy remains near the 50% chance baseline across all tested sequence lengths.

**In-context Learning.** An alternative to fine-tuning for adapting a pre-trained model is in-context learning (ICL). This experiment investigates whether a large, pre-trained LLM can infer the underlying function from examples provided directly in its context window and then apply that inferred rule to classify a new input. Formally, we test if the model's predictive function, $h(x_{\text{test}}|S_{\text{tr}})$, can approximate the target function $y(x_{\text{test}})$, where the training set $S_{\text{tr}}$ (200 samples) is provided as context.

For this evaluation, we employed three large-scale, instruction-tuned models: `Qwen3-30B-A3B-Instruct-2507` (Yang et al., 2025), `Qwen3-Coder-30B -A3B-Instruct` (Yang et al., 2025), and `Deepseek-Coder-33B-Instruct` (Guo et al., 2024). For each of the 100 test samples, a prompt Fig. 11 was constructed containing the problem statement, all 200 training examples, and a single test input, asking the model to predict the corresponding label. Generation was performed with deterministic settings to encourage logical reasoning (temperature of 0.2, top-p of 0.95) and a maximum of 1024 new tokens.

**Results.** The results show the failure of in-context learning to solve these algorithmic tasks. As detailed in the lower half of Tab. 3 and Fig. 10, test accuracy across almost all tasks and sequence lengths consistently hovered around the 50% chance level. This outcome was consistent across all three large models tested. IsPrime, in-context learning yielded minor improvements (e.g., 66% at $n=50$ for `Qwen3-30B-A3B-Instruct-2507`).

---

**LLM Prompt**

**Problem Statement:** Given a sequence of input vectors (binary, length {sequence_dimension}) and their corresponding scalar binary outputs ('0' or '1'), you have to learn a hypothesis that approximates the underlying relationship. Given the data below, determine what is the label for the given string and output ONLY the label. **Data Examples:**

```
000111101011110010100101001100 -> 1
... 011011010111000010010101001000 -> 1
```

**Test Input:**

```
010100110111001001010101001000
```

**You must output ONLY a single JSON object: {"lable": "<your predicted label>"}**

---

Figure 11: Prompt used in in-context learning procedure. We run three models `Qwen3-30B-A3B-Instruct-2507`, `Qwen3-Coder-30B-A3B-Instruct`, and `Deepseek-Coder-33B-Instruct` with this prompt. For each prompt, the model outputs only the predicted label for the test input.

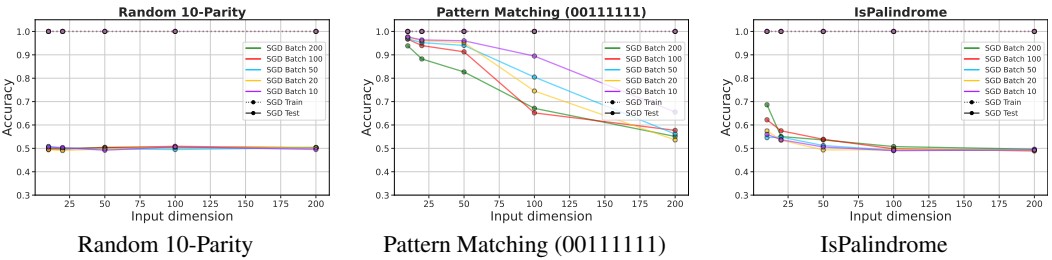

| Random 10-Parity | Pattern Matching (00111111) | IsPalindrome |

Figure 12: **Varying the batch size does not resolve overfitting.** We train instances of `Qwen3-1.7B` on Random 10-Parity **(left)**, Pattern Matching (00111111) **(middle)**, and IsPalindrome **(right)**, reporting train and test accuracy. Changing the batch size does not materially alter the models' tendency to overfit.

### C.2.2 BATCH SIZE

To assess the role of optimization dynamics, we ran a batch-size ablation with 200 training samples and batch sizes of 10, 20, 50, 100, and 200, holding all other hyperparameters fixed.

As shown in Tab. 4 and Fig. 12, generalization is only weakly affected by batch size. Random 10-Parity stays at chance across all settings, while the performance on IsPalindrome and Pattern Matching vary modestly by task. Most importantly, for each task and every batch size, generalization declines sharply as sequence length grows. This suggests the core behavior—memorizing complex patterns while failing to generalize cannot be easily solved by tuning the batch size.

### C.2.3 LEARNING RATE

The learning rate is a critical factor in model convergence and generalization. To examine whether the poor generalization of SGD on the proposed program learning tasks is due to learning-rate choice, we conducted an ablation study. Specifically, we performed a comprehensive sweep across seven orders of magnitude, from $8.0$ down to $8 \times 10^{-7}$. We evaluated `Qwen3 1.7B` on three representative tasks—Random 10-Parity, Pattern Matching (00111111), and IsPalindrome. Each model was trained for 200 epochs with batch size 20 on 200 training samples and evaluated on 10k random test samples.

As shown in Tab. 5 and Fig. 13, the effect of the learning rate varies substantially across tasks. For Random 10-Parity, generalization failure is insensitive to $\eta$: test accuracy remains near 50% across the entire sweep, indicating that no learning rate enables generalization. In contrast, IsPalindrome and Pattern Matching exhibit strong sensitivity to $\eta$. Pattern Matching can be solved perfectly for short sequences ($n$=20), but this success is brittle and does not extend to longer inputs. Similarly, IsPalindrome shows modest generalization for small $n$ only at very low learning rates. For these

| Task | Batch Size | $n = 20$ | $n = 25$ | $n = 30$ | $n = 50$ | $n = 100$ |
|---|---|---|---|---|---|---|
| **Random 10-Parity** | 10 | 50.6% | **50.4%** | 49.2% | 50.7% | 49.5% |
| | 20 | 49.4% | 49.1% | 49.8% | 50.6% | 50.3% |
| | 50 | **50.8%** | 50.0% | 49.7% | 49.5% | 50.3% |
| | 100 | 49.6% | 49.7% | 50.3% | **50.9%** | 50.2% |
| | 200 | 50.1% | 49.8% | **50.4%** | 50.3% | **50.4%** |
| **Pattern Matching (00111111)** | 10 | **97.7%** | **96.4%** | **96.0%** | **89.5%** | **65.5%** |
| | 20 | 97.5% | 96.0% | 95.2% | 74.5% | 53.6% |
| | 50 | 96.9% | 95.2% | 94.0% | 80.5% | 56.1% |
| | 100 | 96.7% | 94.0% | 91.3% | 65.2% | 57.7% |
| | 200 | 93.8% | 88.2% | 82.7% | 67.1% | 55.1% |
| **IsPalindrome** | 10 | 56.0% | 53.7% | 50.6% | 49.0% | 49.3% |
| | 20 | 57.5% | 53.5% | 49.3% | 49.4% | 49.2% |
| | 50 | 54.6% | 54.8% | 51.2% | 49.2% | 49.3% |
| | 100 | 62.2% | **57.5%** | **53.9%** | 49.8% | 48.9% |
| | 200 | **68.6%** | 55.0% | 53.6% | **50.8%** | **49.6%** |

Table 4: **Accuracy is far more sensitive to sequence length than to batch size.** Random 10-Parity stays at chance level across all configurations. Pattern Matching achieves high accuracy on short sequences but performance drops sharply as the sequences grow longer, regardless of batch size. For IsPalindrome, larger batches provide some benefit on shorter sequences, but accuracy still declines toward chance-level on longer ones.

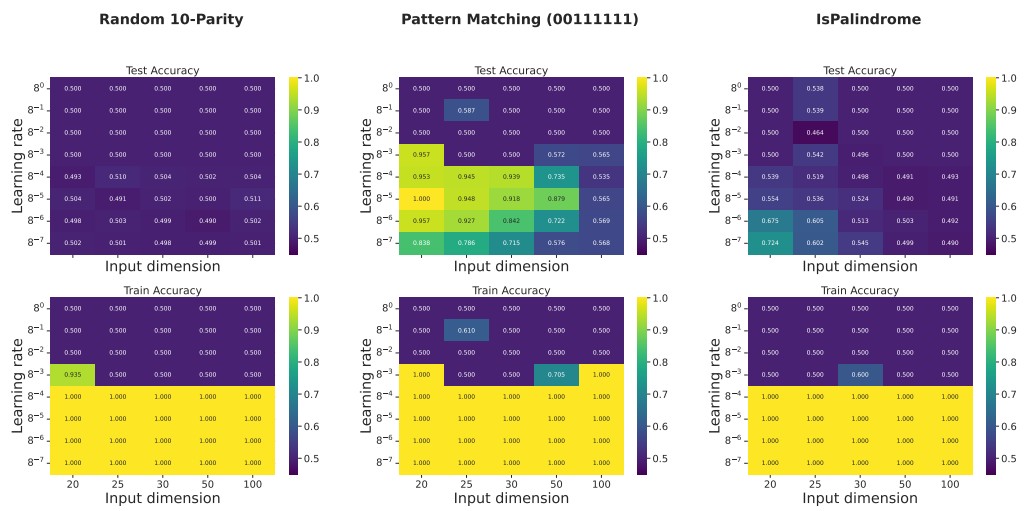

Figure 13: **SGD's failure to generalize is not due to poor learning-rate choices.** We trained Qwen3 1.7B for 200 epochs (batch size 20) with learning rates $\eta$ swept across seven orders of magnitude, reporting train and test accuracy as sequence length $n$ varies. Results are shown for Random 10-Parity **(left)**, Pattern Matching (00111111) **(middle)**, and IsPalindrome **(right)**. For Random 10-Parity, test accuracy remains at chance ($\approx 50\%$) regardless of $\eta$, indicating no learning rate yields generalization. For Pattern Matching and IsPalindrome, certain $\eta$ values succeed on short sequences ($n = 20$) but fail to generalize as $n$ increases.

more complex tasks, no broad optimal range exists; performance is highly sensitive, and effective generalization remains weak, particularly for longer sequences.

## C.3 LLM REASONING TRACES

We extend the reasoning-trace analysis from Fig. 1 to the Cellular Automata Parity and Full Parity tasks, in order to further reveal the model's adaptive problem-solving strategies.

| Task | Learning Rate ($\eta$) | $n = 20$ | $n = 25$ | $n = 30$ | $n = 50$ | $n = 100$ |
|---|---|---|---|---|---|---|
| **Random 10-Parity** | $8 \times 10^0$ | 50.0% | 50.0% | 50.0% | 50.0% | 50.0% |
| | $8 \times 10^{-1}$ | 50.0% | 50.0% | 50.0% | 50.0% | 50.0% |
| | $8 \times 10^{-2}$ | 50.0% | 50.0% | 50.0% | 50.0% | 50.0% |
| | $8 \times 10^{-3}$ | 50.0% | 50.0% | 50.0% | 50.0% | 50.0% |
| | $8 \times 10^{-4}$ | 49.3% | 51.0% | **50.4%** | **50.2%** | 50.4% |
| | $8 \times 10^{-5}$ | **50.4%** | 49.1% | 50.2% | 50.0% | **51.1%** |
| | $8 \times 10^{-6}$ | 49.9% | **50.3%** | 49.9% | 49.0% | 50.2% |
| | $8 \times 10^{-7}$ | 50.2% | 50.1% | 49.8% | 49.9% | 50.1% |
| **Pattern Matching** **(00111111)** | $8 \times 10^0$ | 50.0% | 50.0% | 50.0% | 50.0% | 50.0% |
| | $8 \times 10^{-1}$ | 50.0% | 58.7% | 50.0% | 50.0% | 50.0% |
| | $8 \times 10^{-2}$ | 50.0% | 50.0% | 50.0% | 50.0% | 50.0% |
| | $8 \times 10^{-3}$ | 95.7% | 50.0% | 50.0% | 57.2% | 56.5% |
| | $8 \times 10^{-4}$ | 95.2% | 94.5% | **93.9%** | 73.5% | 53.5% |
| | $8 \times 10^{-5}$ | **100.0%** | **94.8%** | 91.8% | **87.9%** | 56.5% |
| | $8 \times 10^{-6}$ | 95.7% | 92.7% | 84.2% | 72.2% | **56.9%** |
| | $8 \times 10^{-7}$ | 83.8% | 78.6% | 71.5% | 57.6% | 56.8% |
| **IsPalindrome** | $8 \times 10^0$ | 50.0% | 53.8% | 50.0% | 50.0% | **50.0%** |
| | $8 \times 10^{-1}$ | 50.0% | 53.9% | 50.0% | 50.0% | **50.0%** |
| | $8 \times 10^{-2}$ | 50.0% | 46.4% | 50.0% | 50.0% | **50.0%** |
| | $8 \times 10^{-3}$ | 50.0% | 54.2% | 49.6% | 50.0% | **50.0%** |
| | $8 \times 10^{-4}$ | 53.9% | 51.9% | 49.8% | 49.1% | 49.4% |
| | $8 \times 10^{-5}$ | 55.4% | 53.6% | 52.4% | 49.0% | 49.1% |
| | $8 \times 10^{-6}$ | 67.5% | **60.5%** | 51.3% | **50.3%** | 49.2% |
| | $8 \times 10^{-7}$ | **72.4%** | 60.2% | **54.5%** | 49.9% | 49.0% |

Table 5: Test accuracy across a wide range of learning rates ($\eta$). The model's inability to generalize on Parity tasks persists regardless of $\eta$. Pattern Matching and IsPalindrome learns for shorter sequence lengths but accuracy degrades on longer sequence lengths.

| | **BLOOM-75M** | | | | |
|---|---|---|---|---|---|
| Task | $n = 20$ | $n = 25$ | $n = 30$ | $n = 50$ | $n = 100$ |
| **Rand. 10-Parity** | 53.9% | 49.8% | 50.5% | 49.2% | 50.7% |
| **Cellular Automata Parity** | 99.9% | 50.3% | 50.2% | 49.5% | 50.4% |
| **IsPrime (Ends in {1,3,7,9})** | 59.8% | 58.7% | 60.3% | 60.1% | 59.9% |

Table 6: **Test accuracy (%) for `BLOOM-75M` trained on 100k examples per task.** Despite substantially more data, the model overfits and fails to achieve algorithmic generalization: performance is near chance on Random 10-Parity and Cellular Automata Parity across lengths, and only modest on ISPRIME with restricted negatives.

For the more complex Cellular Automata Parity task (Fig. 14), the reasoning trace initially mirrors the approach observed for Random 10-Parity (Fig. 1b). The model begins by testing simple batch statistics and basic parity checks, then searches for a linear solution. When no straightforward rule emerges, it escalates to systematically exploring a richer feature space of non-linear combinations. This includes testing thresholds and computing the parity of masked compositions that involve diverse features such as the first and last bits, a majority-ones flag, the parity of bit flips, and the parity of specific bigrams ("01", "10", etc.). This exhaustive exploration eventually produces the correct, more complex hypothesis. Finally, the model attempts a simplification step before converging on and confirming this solution, indicating a form of internal verification.

In stark contrast, when presented with the Full Parity task, the model's reasoning is immediate and conclusive. As shown in Fig. 15, it quickly dismisses simple heuristics such as ones-count thresholding, then immediately proposes the exact full-parity function over all bits. The hypothesis is

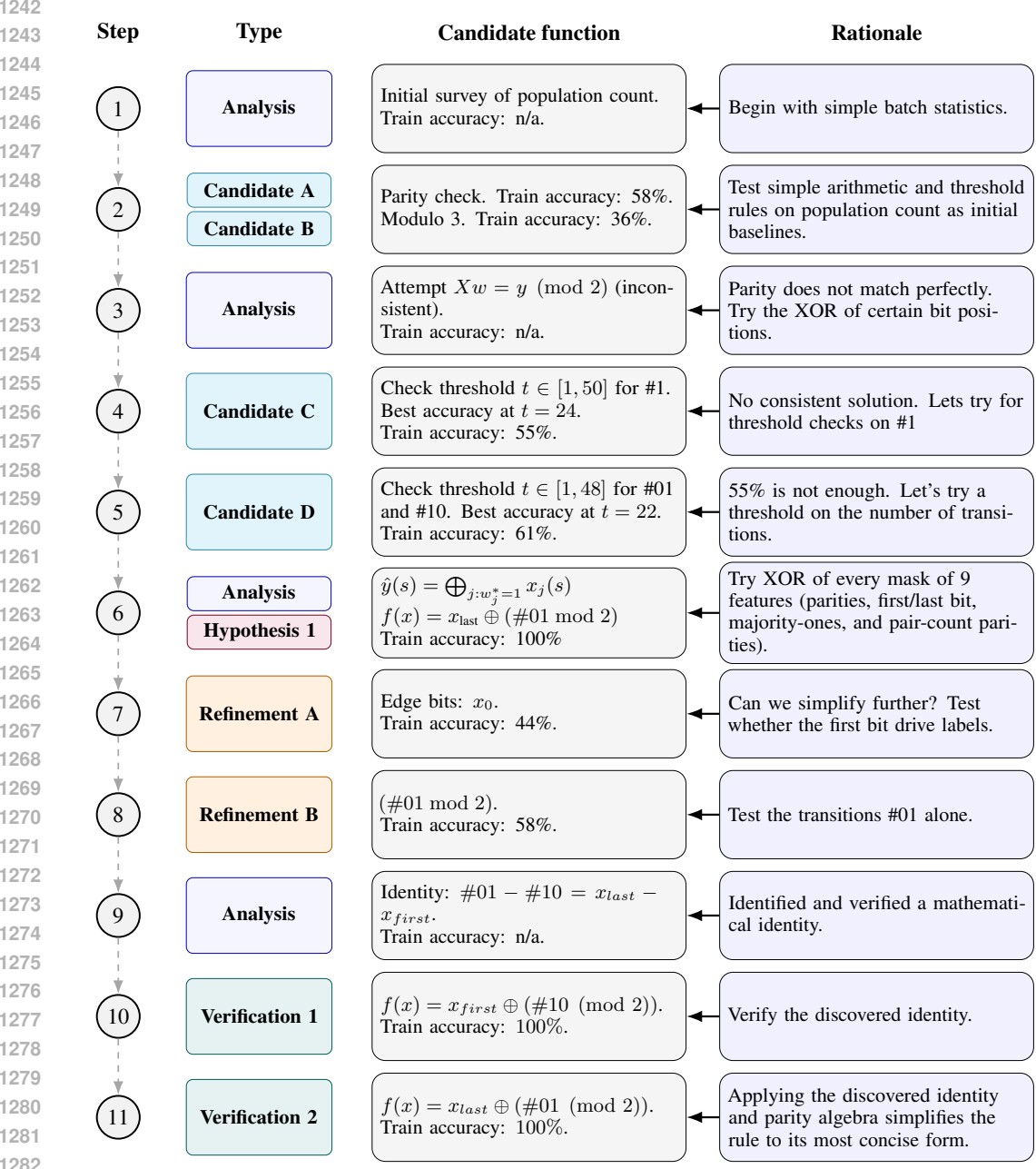

Figure 14: **Reasoning trace for inferring an equivalent rule for Cellular Automata Parity function.** The model starts with simple heuristics, explores linear solutions over $\mathbb{F}_2$, and converges to a global XOR rule that perfectly matches the provided dataset, effectively inferring a simpler, equivalent function.

tested against the dataset, achieves perfect alignment with the labels, and is subsequently verified without the need for extended exploration.

Unlike the Cellular Automata Parity case, where the model incrementally explores a large feature space of non-linear candidates, here the solution emerges almost instantly and is verified in a single step, highlighting the model's ability to identify and lock onto the correct global rule when it is especially simple.

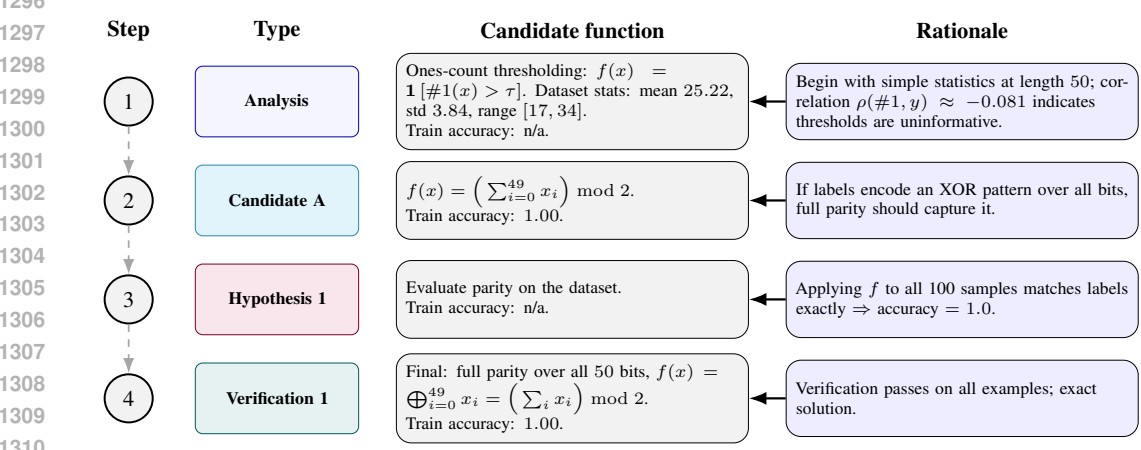

Figure 15: **Reasoning trace for learning a full-parity function.** We list the candidate functions proposed by `GPT-5-thinking` when trained on 100 binary strings of length 50. **(left)** The sequence of candidates explored. **(right)** The rationale for proposing each candidate. The model starts with simple statistics, hypothesizes full parity, and verifies that parity perfectly matches all labels.

---

**Algorithm 2** Length-First Program Search (LFPS)

---

**Require:** Sample $S = \{(x_i, y_i)\}_{i=1}^m$, language $\mathcal{L} \subseteq \Sigma^*$, per-run timeout $T \in \mathbb{N}$, optional max length $L_{\max} \in \mathbb{N} \cup \{\infty\}$
**Ensure:** A program $u^\star \in \mathcal{L}$ whose total semantics $[\![u^\star]\!] : \mathcal{X} \to \{\pm 1\}$ satisfies $[\![u^\star]\!](x_i) = y_i$ for all $(x_i, y_i) \in S$; or $\perp$ if none is found up to $L_{\max}$
  1: **for** $\ell = 1, 2, \dots, L_{\max}$ **do**
  2:   **for all** strings $u \in \mathcal{L}$ with $|u| = \ell$ in lexicographic order **do**
  3:     **if** $u$ fails to compile **then continue**
  4:     $consistent \leftarrow$ true
  5:     **for** each $(x_i, y_i) \in S$ **do**
  6:       Run $u$ on input $x_i$ for at most $T$ steps; let $o_i \in \{\pm 1, \perp\}$ be the output ($\perp$ if no halt)
  7:       **if** $o_i = \perp$ **or** $o_i \neq y_i$ **then**
  8:         $consistent \leftarrow$ false; **break**
  9:       **end if**
 10:     **end for**
 11:     **if** $consistent$ **then**
 12:       **return** $u^\star \leftarrow u$                    {minimal-length consistent program}
 13:     **end if**
 14:   **end for**
 15: **end for**
 16: **return** $\perp$                    {no consistent total program found up to $L_{\max}$}

---

## D    PROOFS

**Theorem 1** (Valiant (1984); see also Cor. 2.3 of Shalev-Shwartz and Ben-David (2014)). *Let* $y : \mathcal{X} \to \{\pm 1\}$ *be an unknown target function and let* $\mathcal{H} \subset \{\pm 1\}^{\mathcal{X}}$ *be a finite hypothesis class. Suppose we are in the realizable setting (i.e.,* $y \in \mathcal{H}$*). Let* $S = \{(x_i, y(x_i))\}_{i=1}^m$ *be* $m$ *training examples drawn i.i.d. from a distribution* $D$ *over* $\mathcal{X} \times \{\pm 1\}$*. Then, with probability at least* $1 - \delta$ *over the draw of* $S$*, every hypothesis* $h \in \mathcal{H}$ *that is consistent with* $S$ *satisfies*

$$\operatorname{err}_D(h) \leq \frac{\log(|\mathcal{H}|) + \log(1/\delta)}{m}.$$

**Corollary 1.** *Let* $y : \mathcal{X} \to \{\pm 1\}$ *be an unknown target function and let* $\mathcal{H} = \bigcup_{\ell \geq 1} \mathcal{H}_\ell \subset \{\pm 1\}^{\mathcal{X}}$ *be a union of finite sets. Suppose we are in the realizable setting (i.e.,* $y \in \mathcal{H}$*). Let* $S = \{(x_i, y(x_i))\}_{i=1}^m$

be $m$ training examples drawn i.i.d. from a distribution $D$ over $\mathcal{X} \times \{\pm 1\}$. Then, with probability at least $1 - \delta$ over the draw of $S$, for any $\ell \in \mathbb{N}$ and any hypothesis $h \in \mathcal{H}_\ell$ that is consistent with $S$, it holds that

$$\mathrm{err}_D(h) \ \leq \ \frac{\log(|\mathcal{H}_\ell|) + \log\big((\pi^2/6)\,\ell^2/\delta\big)}{m}.$$

*Proof.* Assume that $y \in \mathcal{H} = \bigcup_{\ell \geq 1} \mathcal{H}_\ell$; hence, there exists some $\ell^*$ for which $\mathcal{H}_{\ell^*}$ is realizable. For each fixed $\ell$ with at least one hypothesis consistent with $S$, Thm. 1 implies that for any $\delta_\ell > 0$, with probability at least $1 - \delta_\ell$, every $h \in \mathcal{H}_\ell$ consistent with $S$ satisfies

$$\mathrm{err}_D(h) \ \leq \ \frac{\log(|\mathcal{H}_\ell|) + \log(1/\delta_\ell)}{m}.$$

Choose $\delta_\ell = \frac{6}{\pi^2}\,\frac{\delta}{\ell^2}$, so that $\sum_{\ell \geq 1} \delta_\ell = \delta$. Then, for each such $\ell$, with probability at least $1 - \delta_\ell$,

$$\mathrm{err}_D(h) \ \leq \ \frac{\log(|\mathcal{H}_\ell|) + \log\big((\pi^2/6)\,\ell^2/\delta\big)}{m}.$$

Applying a union bound over all $\ell \geq 1$ yields the claim (for each $\ell$ with no consistent hypothesis, the inequality is vacuous). $\qquad\square$

**Proposition 3.** *Suppose we wish to learn a target function $y : \mathcal{X} \to \{\pm 1\}$ that can be implemented as a program of length $L$ in a programming language $\mathcal{L}$. Let $\mathcal{L}_\ell$ denote the set of programs of length $\ell$ in $\mathcal{L}$, and let $S = \{(x_i, y(x_i))\}_{i=1}^m$ be $m$ training examples drawn i.i.d. from a distribution $D$ over $\mathcal{X} \times \{\pm 1\}$. Then, with probability at least $1 - \delta$ over the draw of $S$, Alg. 2 outputs a program $h \in \mathcal{L}$ that is consistent with $S$ and satisfies*

$$\mathrm{err}_D(h) \ \leq \ \frac{L \log |\Sigma| + \log(2L^2/\delta)}{m}.$$

*Proof.* Since $y \in \mathcal{L}$, there exists a minimal length $L$ such that $y \in \mathcal{L}_L$. Therefore, there is at least one program of length $L$ consistent with $S$. Alg. 2 enumerates programs in order of increasing length, so it eventually returns a program $h$ of some length $\ell \leq L$ that is consistent with $S$. Every program in $\mathcal{L}_\ell$ is described over the alphabet $\Sigma$, hence $|\mathcal{L}_\ell| \leq |\Sigma|^\ell$ and $\log |\mathcal{L}_\ell| \leq \ell \log |\Sigma| \leq L \log |\Sigma|$. Applying Cor. 1 with $\mathcal{H} = \mathcal{L}$ and $\mathcal{H}_\ell = \mathcal{L}_\ell$ and then upper-bounding by $L$ gives

$$\mathrm{err}_D(h) \ \leq \ \frac{\log |\mathcal{L}_\ell| + \log(2\ell^2/\delta)}{m} \ \leq \ \frac{L \log |\Sigma| + \log(2L^2/\delta)}{m}.$$

$\qquad\square$

### D.1 FROM MINI-BATCH SGD TO 1-STAT(B) TO VSTAT: A FORMAL REDUCTION

Throughout, $\mathrm{err}_D$ denotes the 0–1 error. All query functions are measurable and bounded.

### D.2 ORACLE MODELS

**Definition 2** (1-STAT and 1-STAT(b))**.** *Let $D$ be a distribution over $\mathcal{X} \times \{\pm 1\}$. A 1-STAT oracle takes a Boolean function $g : \mathcal{X} \times \{\pm 1\} \to \{0, 1\}$, draws a fresh $(x, y) \sim D$, and returns $g(x, y)$. For $b \in \mathbb{N}$, a 1-STAT(b) oracle takes a vector of Boolean functions $g = (g_1, \ldots, g_b)$ and returns the $b$-bit vector $\big(g_1(x, y), \ldots, g_b(x, y)\big)$ for a fresh $(x, y) \sim D$.*

**Definition 3** (VSTAT (Feldman, 2017, Definition 2.3))**.** *Let $D$ be as above. A VSTAT$(t)$ oracle takes $g : \mathcal{X} \times \{\pm 1\} \to [0, 1]$ and returns a value $v \in \mathbb{R}$ such that, writing $p = \mathbb{E}_D[g(x, y)]$,*

$$|v - p| \ \leq \ \max\left\{\tfrac{1}{t}, \ \sqrt{\tfrac{p(1-p)}{t}}\right\}.$$

*(The choice of $v$ within the interval is adversarial; the interval width scales like the standard deviation of $t$ i.i.d. samples.)*

### D.3 SIMULATING MINI-BATCH SGD USING 1-STAT(B)

**Proposition 4** (SGD ⇒ 1-STAT(b))**.** *Assume per-example coordinate gradients are uniformly bounded:*

$$\left| \partial_j \ell(h_\theta, (x, y)) \right| \leq G \qquad \text{for all } \theta, j, (x, y).$$

*At iteration t, define*

$$\phi_t(x, y) := \frac{1}{2} \left( 1 + \frac{1}{G} \, \partial_{j_t} \ell \big( h_{\theta_t}, (x, y) \big) \right) \in [0, 1].$$

*Fix a quantization accuracy $\alpha \in (0, 1)$ and let $b = \lceil \log_2(1/\alpha) \rceil$. Then there exist Boolean functions $g_{t,1}, \ldots, g_{t,b}$ (depending on $t, \theta_t, j_t$) and a deterministic decoder $\mathrm{Dec} : \{0, 1\}^b \to [0, 1]$ such that, for every $(x, y)$,*

$$\big| \mathrm{Dec}\big(g_{t,1}(x, y), \ldots, g_{t,b}(x, y)\big) - \phi_t(x, y) \big| \leq \alpha.$$

*Consequently, the mini-batch average of $\phi_t$ over $B$ i.i.d. samples can be simulated by $B$ calls to 1-STAT(b) at iteration t, with deterministic quantization error at most $\alpha$. Choosing $\alpha = \frac{1}{2\sqrt{B}}$ ensures this quantization error is dominated by the sampling error $O(1/\sqrt{B})$.*

*Proof.* Fix $\alpha \in (0, 1)$. Define the grid

$$\mathcal{G}_b := \left\{ \frac{k}{2^b} : k = 0, 1, \ldots, 2^b \right\} \subseteq [0, 1].$$

Since $b = \lceil \log_2(1/\alpha) \rceil$, the grid spacing is $2^{-b} \leq \alpha$.

Define the quantizer $Q : [0, 1] \to \mathcal{G}_b$ that maps $z \in [0, 1]$ to the unique grid point $Q(z) \in \mathcal{G}_b$ satisfying $|Q(z) - z| \leq 2^{-b} \leq \alpha$. This map can be implemented by encoding the binary expansion of $z$ to $b$ bits, truncated or rounded as needed.

For each $(x, y)$, define

$$(g_{t,1}(x, y), \ldots, g_{t,b}(x, y)) := \text{binary representation of } Q(\phi_t(x, y)).$$

By construction, each $g_{t,i}$ is a Boolean function of $(x, y)$, and

$$\mathrm{Dec}(g_{t,1}(x, y), \ldots, g_{t,b}(x, y)) := Q(\phi_t(x, y)).$$

Therefore

$$\big| \mathrm{Dec}(g_{t,1}(x, y), \ldots, g_{t,b}(x, y)) - \phi_t(x, y) \big| \leq \alpha.$$

Now consider a mini-batch $S = \{(x_1, y_1), \ldots, (x_B, y_B)\}$ of $B$ independent draws from $D$. The SGD update uses the empirical average

$$\hat{v} := \frac{1}{B} \sum_{i=1}^{B} \phi_t(x_i, y_i).$$

Meanwhile, simulating with 1-STAT(b) queries, we obtain

$$\hat{v}_Q := \frac{1}{B} \sum_{i=1}^{B} \mathrm{Dec}(g_{t,1}(x_i, y_i), \ldots, g_{t,b}(x_i, y_i)).$$

For each $i$, the error $|\mathrm{Dec}(g_{t,1}(x_i, y_i), \ldots, g_{t,b}(x_i, y_i)) - \phi_t(x_i, y_i)| \leq \alpha$, hence

$$|\hat{v}_Q - \hat{v}| \leq \frac{1}{B} \sum_{i=1}^{B} \alpha = \alpha.$$

Thus $\hat{v}_Q$ simulates $\hat{v}$ up to additive error $\alpha$. Choosing $\alpha = 1/(2\sqrt{B})$ ensures this error is smaller than the typical sampling deviation of order $1/\sqrt{B}$, so quantization does not alter asymptotics. $\quad\square$

## D.4 SIMULATING 1-STAT(B) BY VSTAT

**Definition 4** (Success predicate). *Let $f^\star \in \mathcal{C}$ be the target and $D$ a distribution over $\mathcal{X}$. Given $\epsilon \in (0, 1/2)$, an algorithm succeeds if it outputs a hypothesis $h : \mathcal{X} \to \{-1, +1\}$ with error $\leq \frac{1}{2} - \epsilon$.*

**Theorem 2** (Feldman et al., 2018, Thm. B.4). *Let $\beta \in (0, 1]$. Suppose there exists an algorithm $\mathcal{A}$ that uses $q$ queries to a 1-STAT(b) oracle and, with probability at least $\beta$, succeeds in the sense of Definition 4. Then for any $\delta \in (0, 1)$ there exists an algorithm $\mathcal{A}'$ that uses at most*

$$Q = \mathcal{O}(q\, 2^b) \text{ queries to } \mathrm{VSTAT}\left(\Theta(q\, 2^b/\delta^2)\right)$$

*and succeeds with probability at least $\beta - \delta$.*

## D.5 FROM VSTAT LOWER BOUNDS TO SGD ITERATION LOWER BOUNDS

**Proposition 2** (Lower bound for SGD). *Let $\mathcal{C}$ be a class with $\mathrm{SQ\text{-}DIM}_D(\mathcal{C}) = d$. Consider coordinate mini-batch SGD with batch size $B$ run for $T$ iterations. Fix $\epsilon \in (0, 1/2)$. If the algorithm outputs a hypothesis of error at most $1/2 - \epsilon$ with probability at least $2/3$, then $T \geq \Omega\left(\frac{d\, \epsilon^2}{B^{3/2}}\right)$.*

*Proof.* We prove the result by reducing any successful run of mini-batch SGD to an algorithm that makes a limited number of queries to a VSTAT oracle. This allows us to invoke standard SQ lower bounds, which force the number of SGD iterations to be large.

**Step 1 (SQ lower bound at accuracy $\Theta(\epsilon)$).** By the standard SQ lower bound for classes with $\mathrm{SQ\text{-}DIM}_D(\mathcal{C}) = d$ (Blum et al., 1994; see also Reyzin, 2020, Theorem 12), any learner that succeeds with error $\leq 1/2 - \epsilon$ with probability $\geq 2/3$ using a $\tau$–tolerant SQ oracle with $\tau = \Theta(\epsilon)$ must make at least $Q^\star = \Omega(d\,\epsilon^2)$ SQ queries. Equivalently, since a single SQ query of tolerance $\tau$ can be answered by one VSTAT($t$) query with $t = \Theta(1/\tau^2)$ (and vice versa), the same lower bound holds for VSTAT($t^\star$) with

$$t^\star = \Theta(1/\epsilon^2): \qquad \text{any VSTAT}(t^\star) \text{ learner that succeeds must use} \geq Q^\star \text{ queries.} \qquad (\dagger)$$

**Step 2 (Express one SGD step via 1-STAT and then via VSTAT).** By Proposition 4, one iteration of coordinate mini-batch SGD with batch size $B$ can be simulated by $B$ queries to a 1-STAT($b$) oracle, with $b = \lceil \log_2(2\sqrt{B}) \rceil$. Hence the full run (over $T$ iterations) uses $q = TB$ queries to 1-STAT($b$). By Thm. 2, for any fixed $\delta \in (0, 1/6)$, there is a transformation that simulates this 1-STAT($b$) algorithm by an algorithm that makes $Q_0 = O(q\, 2^b) = O(TB\, 2^b)$ queries to a VSTAT($t_0$) oracle (for some $t_0 = \Theta(q\, 2^b/\delta^2)$), and succeeds with probability at least $2/3 - \delta$.

**Step 3 (Amplify success probability to $\geq 2/3$).** Set $\delta = 1/12$. The simulated VSTAT($t_0$) algorithm from Step 2 succeeds with probability $p_0 = 2/3 - \delta = 7/12$. Run $r$ independent copies to obtain hypotheses $h_1, \ldots, h_r$. For each $j$, estimate the error $e_j := \mathrm{err}_D(h_j)$ using one VSTAT($t_{\mathrm{sel}}$) query on $h'_j(x, y) = \mathbf{1}[h_j(x) \neq y]$, with $t_{\mathrm{sel}} = \Theta(1/\epsilon^2)$, which returns $\hat{e}_j$ satisfying $|\hat{e}_j - e_j| \leq \epsilon/4$. Output $h_\star = \arg\min_j \hat{e}_j$.

With probability $1 - (1 - p_0)^r$ at least one copy is $\epsilon$–good (i.e., has $e_j \leq \frac{1}{2} - \epsilon$). On that event, the selection rule guarantees

$$e_\star \leq \min_j e_j + \tfrac{\epsilon}{2} \leq \tfrac{1}{2} - \epsilon + \tfrac{\epsilon}{2} = \tfrac{1}{2} - \tfrac{\epsilon}{2}.$$

Taking $r = 3$ gives $1 - (1 - p_0)^3 = 1 - (5/12)^3 > 2/3$. Thus, after a constant number of repetitions and a constant number of additional VSTAT queries (for selection), we obtain a hypothesis with error at most $\frac{1}{2} - \frac{\epsilon}{2}$ with probability $> 0.9$. Absorbing the constant factor loss in $\epsilon$ into the big-$\Theta$ notation, this yields success probability $> 2/3$ while multiplying the total number of VSTAT queries only by a universal constant, so $Q = \Theta(Q_0) = \Theta(TB\, 2^b)$.

**Step 4 (Align the oracle parameter to $t^\star = \Theta(1/\epsilon^2)$).** The simulation in Step 2 produces a VSTAT($t_0$) oracle with parameter $t_0$ that may differ from $t^\star$. We consider two cases:

*Case 1: $t_0 \geq t^\star$ (more accurate oracle).* A VSTAT($t_0$) reply is guaranteed to be closer to the true expectation than a VSTAT($t^\star$) reply. By post-processing (adding extra random noise), we can

make each VSTAT$(t_0)$ answer distributed exactly as a VSTAT$(t^\star)$ answer, without using additional queries. Thus any algorithm using $Q$ queries to VSTAT$(t_0)$ can be viewed as an algorithm using $Q$ queries to VSTAT$(t^\star)$.

*Case 2: $t_0 < t^\star$ (less accurate oracle).* Suppose, for contradiction, that there exists an algorithm that succeeds with fewer than $Q^\star$ queries to VSTAT$(t_0)$. Since VSTAT$(t^\star)$ is strictly more accurate, the same algorithm would also succeed with the same number of queries to VSTAT$(t^\star)$ (simply by treating each VSTAT$(t^\star)$ answer as a VSTAT$(t_0)$ answer). This contradicts (†).

In either case, success with at most $Q = \Theta(TB\,2^b)$ VSTAT calls would contradict (†) unless $Q \geq Q^\star = \Omega(d\,\epsilon^2)$.

**Step 5 (Conclude and simplify).** We have $\Theta(TB\,2^b) \geq \Omega(d\,\epsilon^2)$, i.e. $T \geq \Omega\left(\frac{d\,\epsilon^2}{B\,2^b}\right)$. Finally, with $b = \lceil \log_2(2\sqrt{B}) \rceil$ we have $2^b = \Theta(\sqrt{B})$, so $T \geq \Omega\left(\frac{d\,\epsilon^2}{B^{3/2}}\right)$, as claimed. $\qquad\square$

