# OpenReview forum: "LLM-ERM: Sample-Efficient Program Learning via LLM-Guided Search"
_ICLR.cc/2026/Conference — ICLR 2026 Conference Withdrawn Submission_

### Official Review · Reviewer_ypu1 · 2025-10-30

**Soundness:** 1
**Presentation:** 2
**Contribution:** 1
**Rating:** 0
**Confidence:** 4

**Summary:**

The authors propose LLM-ERM, an algorithm that samples and validates programs from LLMs given descriptions of inductive programming tasks. They show that LLMs are effective that learning popular classification problems from examples.

The authors also include a broad description of SGD, how it relates to Statistical Queries, and the appendix contains many theorems and proofs about SGD, but they seem only marginally related to the actually proposed algorithm.

**Strengths:**

Learning programs from examples is a challenging and interesting problem, with many real-world applications.

**Weaknesses:**

* This paper is bloated with different theorems and proofs that are completely unrelated to the actually proposed algorithm. The main algorithm—sampling and verifying candidate programs from an LLM—has nothing to do with SGD. Whereas it is true that you could use SGD to optimize a neural network to directly predict the outputs from inputs in inductive programming tasks [1, 2, and many other missing citations] this is a completely different problem than training (or prompting) a neural network to output a program that performs the task.
* The authors claim a "search" but each sample is independent of the previous sample. Strategies like self-debug [3], tree-of-thoughts and execution-guided decoding [4] actually perform a search.
* The authors only evaluate on popular problems, which the model has likely seen the solution of. Only a few out of those 100 provided samples needs to get the model to make the right "click" to suggest a program. I'd be more interested in results on "new" problems (or strongly adapted problems).
* The baselines of fine-tuning smaller transformers on the same amount of samples are very weak, especially since the problems themselves have very complicated decision surfaces (like prime testing). I don't think anyone would even expect this to work. Instead, baselines should explore different LLM-augmented strategies (like the ones mentioned above).
* There are many missing references on (neural) program synthesis and (neural) program induction—see below and many more.

[1] Graves, A., Wayne, G., & Danihelka, I. (2014). Neural turing machines. arXiv preprint arXiv:1410.5401.

[2] Devlin, J., Uesato, J., Bhupatiraju, S., Singh, R., Mohamed, A. & Kohli, P.. (2017). RobustFill: Neural Program Learning under Noisy I/O. Proceedings of the 34th International Conference on Machine Learning, in Proceedings of Machine Learning Research 70:990-998.

[3] Chen, X., Lin, M., Schärli, N., & Zhou, D. Teaching Large Language Models to Self-Debug. In The Twelfth International Conference on Learning Representations.

[4] Verbruggen, G., Tiwari, A., Singh, M., Le, V., & Gulwani, S. Execution-guided within-prompt search for programming-by-example. In The Thirteenth International Conference on Learning Representations.

**Questions:**

* Did you try experiments on new / unknown problems?
* If the solution is just generate-and-test, they should be fit to compute a pass@k rate. What's the pass@1 rate here? As in, how many samples should you draw before finding a correct solution?
* How does searching for a program that minimizes the error help, if there's no way for one program to influence the next one and a program can only be correct if it has zero error?
* The reasoning traces (like Figure 1) are just visualizations of the internal GPT-5 reasoning? I'd be more interested in seeing the raw reasoning and not a human re-interpretation.

---

### Official Review · Reviewer_jUp7 · 2025-10-31

**Soundness:** 2
**Presentation:** 2
**Contribution:** 1
**Rating:** 2
**Confidence:** 4

**Summary:**

The authors propose a propose-and-verify framework that replaces exhaustive enumeration with an LLM-guided search over candidate
programs while retaining ERM-style selection on held-out data. A reasoning augmented pretrained LLM selects k candidates which are then compiled and checked on the data and the best verfieid hypothesis is returned. This approach performs much better (as expected) than traditional SGD based optimization on a given training data

**Strengths:**

-

**Weaknesses:**

The main issue of the paper the lack of novelty and core fundamental contribution. The whole setup and conclusions of the paper is entirely expected. It is well known and very obvious that for simple problems like IsPalindrome, Is Prime, LLM based solutions will perform much better than the  more traditional PAC Learning and empirical risk minimization.

The target tasks considered here are way too simple.
These also being very standard tasks (Full Parity, Is Palindrome) it is entirely expected that LLMs will perform much better and in fact these LLMs have been explicitly trained to understand these very basic concepts

Overall I feel such simplistic tasks are obsolete in the context of LLMs. If LLM Guided Search really has to be evaluated for PAC Learning, it has to be done on more complex, realistic tasks in  a very domain specific setting. For these tasks the performance achieved by the LLM is near to perfect, as is expected and I don’t think there is any novelty in just applying LLM for such simple traditional tasks.

**Questions:**

-

---

### Official Review · Reviewer_EJo7 · 2025-10-31

**Soundness:** 2
**Presentation:** 3
**Contribution:** 2
**Rating:** 2
**Confidence:** 4

**Summary:**

The paper "LLM-ERM: SAMPLE-EFFICIENT PROGRAM LEARNING VIA LLM-GUIDED SEARCH" is an interesting paper that uses language models to solve the following classic problem: given a bunch of iid binary examples x_i, y_i, find a (short) program such that y = f(x) on future examples from the same distribution. It has both theoretical and experimental parts.

I lean towards not accepting this paper because I don't feel that its theoretical or empirical results are strong enough.

I give a 2 for technical soundness since I don't understand some things, but will consider raising my score if they are explained.

**Strengths:**

1. In comparison to all the papers which are doing old-fashioned ML without using LLMs, this paper is more relevant. :-)
2. The problem studied is a classic problem.
3. The paper is pretty well written.
4. Using GPT5, they are able to find programs that would it have been unthinkable to try to find previously.

**Weaknesses:**

1. The lower-bounds are not compared to the very related lower bounds in the (deep) Abbe et al 2021 paper. The theory in this paper is intuitive. The whole point of SQ learning was to show that certain problems like parity can't be learned by algorithms like gradient descent. The Abbe et al 2021 paper went in the surprising direction, showing that *exact* SGD can learn things like parity, which was a big surprise. The Abbe paper also had lower bounds, and showed that SGD can learn parity depending ont the minibatch size and precision. Abbe et al's results aren't discussed in this paper (just cited once) and if this paper does improve on Abbe et al's lower bounds, they need to be compared.
2. There may be a technical flaw in the lower-bound, or at least I didn't understand how the lower-bound argument works. It seems like the logic is: (a) minibatch gradients are a type of low-tolerance oracle (agreed) that you could throw in an SQ SGD algorithm, and (b) any SQ algorithm provably require exponentially many queries for functions like parity. However, the subtly is that the SQ lower-bounds prove that there exists some oracle for which the SQ algorithm fails. In fact, the Abbe et al result shows exactly this, that minibatch SGD can be used to learn parity, depending on the minibatch size and precision. The current paper doesn't even seem to get into precision of SGD, which seems inherent to the delicate argument of Abbe but absent here. In particular, the lower bound from Theorem 1c (Abbe et al 2021 [arxiv version](https://arxiv.org/pdf/2108.04190) ) relies on Lemma 4 (proof in Appendix D) and Theorem 2c (Appendix A.3) and I just don't see how the current paper achieves something similar without that machinery.
3. I didn't understand where the "guided search" is happening and the term loop is also used heavily. I only see a prompt for finding a program, no prompt for improving programs based on what was found thus far. The algorithm appears to be a "best of n" algorithm rather than some fancy guilded search or iterative improvement. Am I missing something?
4. Even if this is all correct, the main point of the paper is kind of obvious: if any college CS student was tasked with finding a program mapping each $x_i$ to $y_i$ the first thing they would do is ask chatGPT. And sure, if it gave a short program then occam bounds would guarantee generalization. The fact that SGD can't solve the problem was already shown by Abbe et al (2021).
5. It would be helpful if the authors better motivate why this problem is still important now in the LLM era, now that we have models like GPT5 which can do very advanced things way beyond the expectations of researchers when they previously worked on program learning.

**Questions:**

1. How does this go beyond Abbe et al (2021)?
2. How do you address my technical flaw, number 2 above?
3. Why is this problem important in the LLM era? Can you revisit it's importance? (I personally don't think that just because it's a classic problem and GPT-5 works way better than anything previously, it's worth writing a paper.)

---

### Official Review · Reviewer_6DRS · 2025-11-02

**Soundness:** 3
**Presentation:** 3
**Contribution:** 2
**Rating:** 2
**Confidence:** 3

**Summary:**

This paper proposes LLM-ERM, a tool that generates an ERM program to solve a given training dataset. It proposes learning a program (here a statistical query) by posing it as a program synthesis problem. They establish theoretically the tradeoffs of SGD v/s enumerating SQs, and propose the LLM-ERM algorithm, which is a propose-and-verify technique that tries to iteratively propose solutions, which are then verified by some oracle.

**Strengths:**

1. Strong theoretical foundations: The authors make good theoretical contributions, analyzing the complexity of solving a problem both by SGD and by SQs. I really enjoyed reading the theoretical analysis.
2. Diverse tasks for evaluation: I appreciated the diversity of the tasks for evaluation. While I do think the tasks themselves are a bit narrow (more in weaknesses), they're still pretty complex depending on the length of the input itself, and non-trivial for an LLM to generate a solution for.

**Weaknesses:**

1. Algorithm: The LLM-ERM technique itself seems quite simple, and honestly, not a particularly significant contribution unless I am missing something really crucial. If I understand Algo 1 correctly, it is simply sampling b programs given an input prompt, checking the error rate of each program, and returning the program if the error rate falls below a threshold. I don't think it is even necessary to iterate $k$ times, one could just sample up to $bk$ samples in one go, and then check each sample. It should not massively increase the latency, since it would be parallelized over the GPU, and would probably be faster than iteratively calling the LLM $k$ times, sampling $b$ outputs each time. I guess the iteration only makes sense if you are using some feedback from iteration $i$ to influence the iteration $i+1$, but I don't see evidence of that. This is then equivalent to measuring pass@$bk$, except instead of finding the program that perfectly matches the input/output, we find the program whose error rate is below the threshold. Also, how do you determine the threshold? I will be willing to increase my score if you can explain why your algorithm is a crucial and novel contribution.

2. Further discussions on the theoretical insights: I really enjoyed reading the theoretical analysis (I still might not understand it entirely), but I feel like there was a lack of followup from the final insight from section 2. While it is nice to get actual bounds on the number of samples and the learning complexity for both approaches, it seems a bit orthogonal from the technique itself, which hinges more on the final two lines (L262--264), which is something kind-of obvious in the program synthesis community. I frankly wanted more: maybe an exploration into how this bound may be pushed down even more, maybe seeing how feedback from the oracle would impact the subsequent iterations, or an additional discussion about the features of problems that would benefit from being framed this way (keeping in mind the side effects that LLMs have with large contexts)

**Questions:**

See weaknesses.

Also:
1. I might have missed it, but how many program samples are evaluated before the LLM-ERM algorithm converges? What is the improvement from LLM-ERM compared to simply asking GPT-5 to produce code to solve the training samples?
2. In the theoretical analysis, you mention complexities for the number of required samples. Can you illustrate how this looks wrt one of the benchmark tasks you evaluate over with concrete numbers? I see the same number of training samples used for each task, though I would assume each task would require a different number of training samples.

---

### Note · Authors · 2025-11-20

I have read and agree with the venue's withdrawal policy on behalf of myself and my co-authors.